# Constructing flexible sub-nanometer ferroelectric catalyst to overcome heterocatalytic kinetic barriers for enhanced catalytic and immuno-therapy

Rui Zhang[1], Yuanfei Yao[2], Lu Yang[1] ✉, Xudong Zhao[1], Boshi Tian[1], Dan Yang[1] ✉ & Piaoping Yang ◉[1] ✉

Piezoelectric catalysis enhances therapeutic outcomes in nanocatalysis but is limited by intrinsic catalysis mechanism. This study employs sub-nanometer $Hf_{0.5}Zr_{0.5}O_2$ (HZO) nanowires as a piezoelectric catalyst to address these challenges. Oxygen K-edge X-ray absorption spectroscopy and spherical aberration-corrected transmission electron microscopy reveal the orthorhombic phase (*Pca21*) in HZO nanowires. This structure imparts polymer-like flexibility to $Hf_{0.5}Zr_{0.5}O_2$, improving its sensitivity to mechanical stress. Molecular dynamics and first-principles calculations demonstrate that ultrasonic stimulation increases the mobility of oxygen bridges, facilitating efficient ferroelectric polarization reversal. This mechanism breaks the "scaling relationship" between the low activation energy for reactant adsorption and the high activation energy for product desorption, enabling significant hydroxyl radical generation. Additionally, hydrogen produced during catalysis promotes pyroptosis, enhancing CD8[+] T cell infiltration and reversing tumor immunosuppression. This research underscores the potential of sub-nanoscale ferroelectric materials in anti-tumor applications.

Piezoelectric catalytic therapy is a significant advancement in nanocatalytic therapy, utilizing the unique properties of piezoelectric materials to enhance catalytic reactions while reducing drug toxicity[1–4]. This therapy generates reactive oxygen species (ROS) under mechanical stimulation, selectively targeting tumor cells[5–8]. Traditional catalysts are limited by the "scaling relationship," which hinders efficient reactant adsorption and product desorption[9–11]. Tuning intermediate binding energy through ferroelectric polarization switching may addresses this challenge, making catalyst performance critical for therapeutic efficacy[12–14].

Inorganic piezoelectric catalysts have the advantage of converting mechanical energy into electrical energy to promote charge separation but often lack stress sensitivity. Conversely, organic materials, though flexible, exhibit poor catalytic performance due to low piezoelectric coefficients[15–17]. Therefore, optimizing catalyst performance involves enhancing the flexibility of inorganic materials and increasing the piezoelectric coefficients of organics[18–20]. Wang et al. proposed that adding polymer-like flexibility to inorganic sub-nanowires could overcome traditional limitations[21–23].

Hafnium dioxide ($HfO_2$) is attractive in biomedicine for its biocompatibility and minimal toxicity, gaining FDA approval as a therapeutic agent[24–26]. Its semiconductor and ferroelectric properties confer piezoelectric catalytic activity[27,28]. Synthesizing $HfO_2$ into sub-nanometer structures could enhance responsiveness to mechanical

[1]Key Laboratory of Superlight Materials and Surface Technology, Ministry of Education, College of Materials Science and Chemical Engineering, Harbin Engineering University, Harbin, PR China. [2]Department of Gastrointestinal Medical Oncology, Harbin Medical University Cancer Hospital, Harbin, PR China. ✉e-mail: yanglu@hrbeu.edu.cn; yangdan@hrbeu.edu.cn; yangpiaoping@hrbeu.edu.cn

vibrations, while its weak ferroelectric domains compromise ferroelectric stability, leading to fluctuating ferroelectric reversal reliability[29–32]. Incorporating zirconium (Zr), also approved by FDA for clinic use, can improve HfO₂'s ferroelectric characteristics[33,34]. This doped structure reduces dipole reversal barriers, enhancing ferroelectric reliability and allowing polarization switching without external fields[35]. This approach can disrupt the scaling relationship in piezoelectric catalysis, facilitating reactant adsorption and product desorption[36–38]. Moreover, $ZrO_2$ can act similarly to metalloenzymes, effectively decomposing hydrogen peroxide ($H_2O_2$) in tumor microenvironments[39–42].

This study explores the potential of sub-nanometer $Hf_{0.5}Zr_{0.5}O_2$ nanowires (HZO NWs) as piezoelectric catalysts for cancer treatment. With polymer-like flexibility, these nanowires enhance sensitivity to ultrasound, leading to polarization reversal and improved catalytic efficiency. They generate significant ROS and hydrogen, while functionalization with DSPE-PEG-TPP targets mitochondrial membranes, triggering pyroptosis and enhancing T cell infiltration in the tumor microenvironment. The non-Fenton decomposition of $H_2O_2$ provides oxygen for therapy, synergizing with immune responses to effectively damage tumor cells (see Fig. 1).

## Results

### Structural characterizations

$Hf_{0.5}Zr_{0.5}O_2$ ultrafine NWs were synthesized via a thermal cracking method involving hafnium and zirconium acetylacetonates in an organic solvent with lithium hydroxide. The reaction was conducted under vacuum at 100 °C for 30 min, followed by argon-filled heating at 300 °C for 3 h to produce HZO NWs. Concurrently, HfO₂ (HO) NWs, ZrO₂ (ZO) NWs, and HZO nanoparticles (HZO NPs) were synthesized using the same method. X-ray diffraction (XRD) analysis (Fig. 2a)

confirmed that HO NWs and ZO NWs were orthorhombic and closely matched standard patterns. Overlapping XRD peaks indicated a structural compatibility between Hf and Zr. The Rietveld refinement of the XRD pattern indicates that the characteristic diffraction peaks for HZO NWs at 23°, 30.20°, 50.09°, and 60.66°, corresponding to (210), (111), (220), (022), and (131) facets of orthogonal HZO (Supplementary Fig. 1)[32].

Transmission electron microscopy (TEM) images indicated that the NWs were homogeneous with ~1 nm in diameter (Fig. 2b–d). High-resolution TEM (HRTEM) showed facet spacings of 0.289 nm for HO NWs and ZO NWs, and 0.293 nm for HZO NWs, all corresponding to (111) facets, confirming the orthorhombic phase structure. Selected area electron diffraction (SAED) patterns exhibited weak diffraction due to the sub-nanometer size of the NWs. Spherical aberration corrected transmission electron microscope (AC-TEM) imaging of HZO NWs (Fig. 2e) further clarified the atomic structure, revealing two symmetrically non-equivalent oxygen atoms with significant deviations from the Hf/Zr atom positions, corroborating the orthorhombic structure.

In contrast, HZO NPs displayed characteristics of the cubic phase (*Fm-3m*) in their XRD patterns (Fig. 2a). TEM images demonstrated their uniform dispersion (Supplementary Fig. 2). The close structural similarity between the cubic and orthorhombic phases posed challenges for differentiation *via* XRD alone. Thus, soft X-ray absorption spectroscopy (SXAS) was employed to analyze orbital components, particularly the O K-edge spectra (Fig. 2f)[43–45]. HZO NPs exhibited $t_2 - e$ energy level splitting due to tetrahedral distortion, while the orthorhombic phase showed additional e level splitting from rhombic cone distortion. The spectral features in the O K-edge X-ray absorption spectroscopy were attributed to crystal field effects from the d0 electron configuration of $Hf^{4+}$ and $Zr^{4+}$, affirming the enhanced

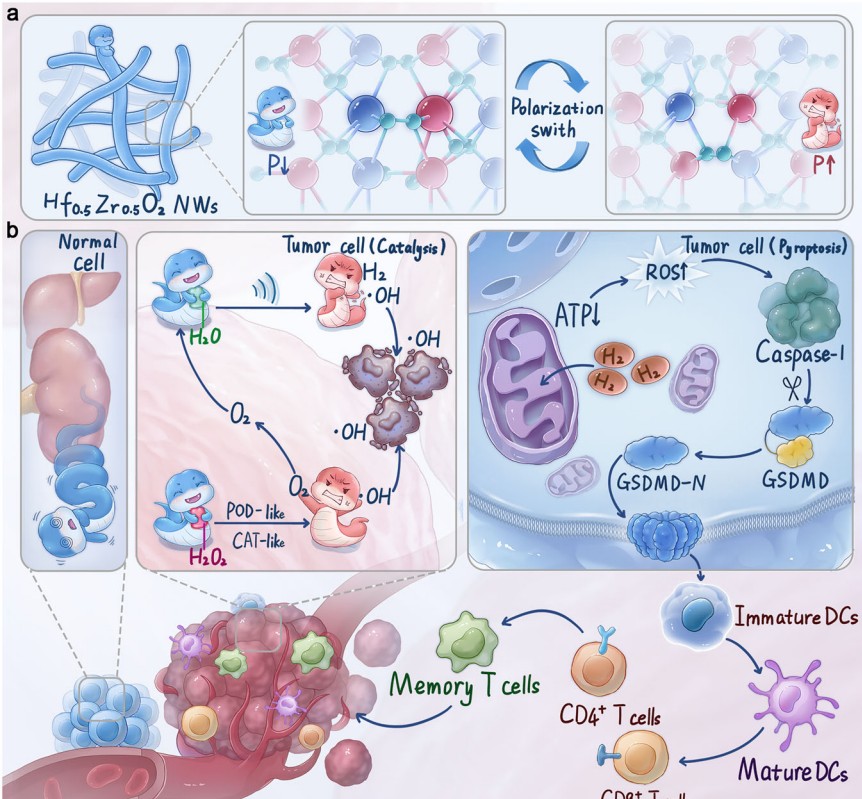

**Fig. 1 | Schematic representation of the structure and synergistic treatment of HZO NWs. a** Structure of sub-nanometer-sized HZO NWs before and after polarization reversal. **b** Under US irradiation, HZO NWs piezoelectrically catalyzed $H_2O$ and $H_2O_2$, which effectively and synergistically inhibited tumor growth and induced cell pyroptosis.

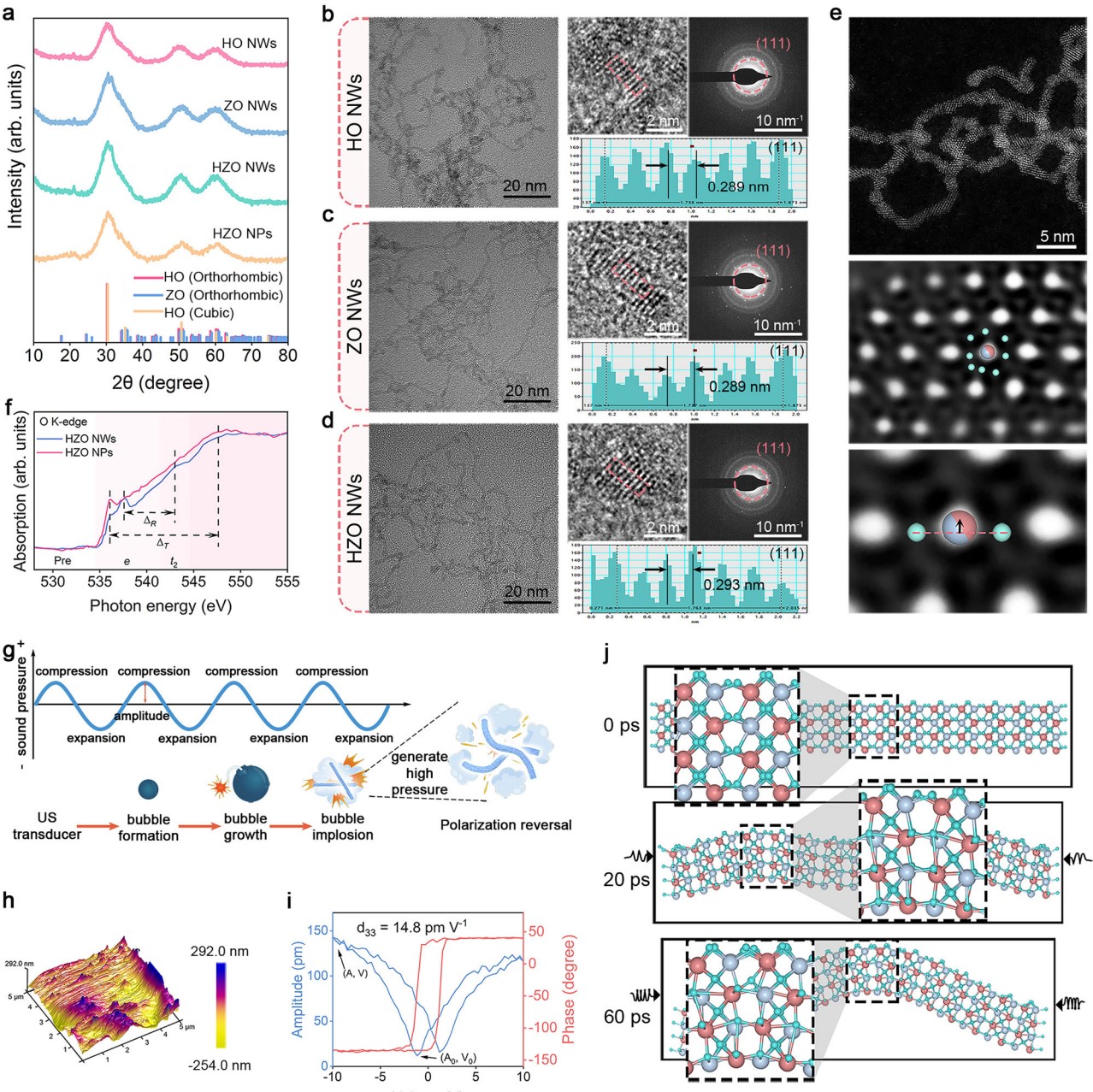

**Fig. 2 | Characterizations and ferroelectric polarization of HZO NWs. a** XRD patterns of HO NWs, ZO NWs, HZO NWs, and HZO NPs. Low- and high-magnification TEM images, SAED pattern, and lattice spacing measurement of **b** HO NWs, **c** ZO NWs, **d** HZO NWs. **e** Atomically resolved AC-TEM image and its enlarged image of HZO NWs. **f** Oxygen K-edge XAS of the HZO NWs and HZO NPs.

**g** Schematic diagram of ferroelectric polarization switching caused by ultrasonic cavitation. **h, i** Piezoresponsive in-plane amplitude, phase curve (red), and amplitude curve (blue). **j** Molecular dynamics simulation under different pressure environments. The illustration is an enlarged image with black frame.

ferroelectric properties of HZO NWs. Zr stabilizes metastable ferroelectric phases in HfO₂, improving properties, and surface modification of HZO NWs enables water dispersion and mitochondrial targeting for applications (Supplementary Discussion 2.1 and Supplementary Figs. 3–11).

### Electron transfer performance

TPP-HZO (THZO) NWs display a high sensitivity to external perturbations due to their small diameter and flexible bending properties, with small vibrations capable of rapidly altering their morphology[46,47]. To investigate this effect, we subjected THZO NWs to ultrasonic treatment. The ultrasonic waves induce alternating positive and negative

pressure within the liquid medium, leading to cyclic compression and expansion (Fig. 2g). Under negative pressure, the liquid experiences tensile stress, resulting in the formation of micro-cavities that expand into bubbles. The cyclic nature of this process causes the bubbles to continuously expand, eventually imploding, releasing a large amount of energy and generating extremely high pressure. This ultrasonic cavitation results in bending of the THZO NWs, which modifies the positions of oxygen atoms and the electron density surrounding the metal elements. Consequently, this activates ferroelectric polarization switching, enhancing electron transfer during catalytic processes.

A ferroelectric hysteresis loop test was conducted to confirm the ferroelectric nature of THZO NWs, revealing a significant residual

polarization (Pr) of 1.45 μC cm$^{-2}$ at 30 kV (Supplementary Fig. 12). This confirms the ferroelectric characteristics of THZO NWs and suggests an asymmetric structure within the *Pca2₁* space group. In contrast, the Pr values for TPP-HfO$_2$ (THO) NWs, TPP-ZrO$_2$ (TZO) NWs, and HZO (HZO) NPs were much lower, at 0.50, 0.26, and 0.17 μC cm$^{-2}$, respectively. These results demonstrate that while orthorhombic HfO$_2$ primarily contributes to ferroelectricity, the coexistence of HfO$_2$ and ZrO$_2$ effectively enhances this property. Further investigations using piezoresponse force microscopy (PFM) revealed remarkable piezoelectric characteristics for THZO NWs when a voltage of 10 V was applied (Fig. 2h). Notably, the PFM phase hysteresis exhibited a 180° change in voltage from −10 to +10 V (Fig. 2i), reaffirming the enhanced ferroelectric properties due to the coexistence of HfO$_2$ and ZrO$_2$ (Supplementary Fig. 13).

To further explore the relationship between polarization reversal and piezoelectric catalytic mechanisms, molecular dynamics simulations were utilized to examine vibration-induced structural strains and the resulting polarization-switching phenomena. As depicted in Fig. 2j, applying gradually increasing pressure to downward polarized THZO NWs induced observable lattice distortion, sufficient to switch the polarization state. The sensitivity of the oxygen lattice arrangement in THZO NWs means that lattice distortion can lead to shifts in oxygen atom positions, directly affecting polarization direction. Our simulations demonstrated that minimal vibrations could trigger polarization switching in HZO NWs (Supplementary Movie). This behavior suggests that THZO NWs maintain their ferroelectric properties while producing surfaces capable of polarization switching. The ability to switch ferroelectric polarization in situ under ultrasonic vibration generates reversible piezoelectric potential, indicating a potential breakthrough in traditional hetero-catalytic kinetics and offering a more direct approach for piezoelectric catalysis in tumor therapy.

Optical characterization of THZO NWs was performed through UV-vis diffuse reflection spectroscopy, estimating an optical band gap of 2.82 eV via the Kubelka-Munk method (Fig. 3a, b). Additionally, density functional theory (DFT) calculations established HZO as a direct bandgap semiconductor with a bandgap of 1.65 eV (Fig. 3c). Using the Mott-Schottky method, we determined the flat band potential ($E_{fb}$(vs Ag/AgCl)) of the THZO NW, which is helpful for deriving the corresponding VB position. The negative slope of the M-S graph represents the typical P-type semiconductor behavior. For the THZO NW with Ag/AgCl electrodes, the $E_{fb}$(vs Ag/AgCl) is 1.63 V (Fig. 3d). The Nernst equation provides 2.24 V relative to the standard hydrogen electrode ($E_{fb}$ (vs NHE)). Furthermore, the valence band energy level ($E_{VB}$) can be calculated as 2.54 V. According to the principle of semiconductor band structure, the conduction band energy level ($E_{CB}$) is calculated to be −0.35 V. However, ultrasonic stress alters the local dipole moment of THZO NWs, generating ferroelectric polarization and establishing a built-in piezoelectric field. This field bends the energy band structure, adjusting the conduction band potential to facilitate electron and hole participation in redox reactions, thereby promoting water splitting into H$_2$ and ·OH (Fig. 3e). Electrochemical analyses of THZO NWs under ultrasonic conditions (Fig. 3f, g) confirmed enhanced separation and transfer of surface carriers under ultrasonic excitation, further supporting the proposed catalytic applications of THZO NWs.

## DFT calculations

To elucidate the kinetic electron transfer mechanisms during the ferroelectric-catalyzed process, we performed first-principles calculations using density functional theory (DFT) implemented in the Vienna Ab initio Simulation Package (VASP). Motivated by molecular dynamics simulations, we examined the displacement of oxygen atoms in THZO NWs under external vibrations, resulting in two polarization transitions: downward (p↓) in the natural state and upward (p↑) under

ultrasonic (US) vibrations (Fig. 3h). The continuous distortion from US vibrations facilitates spontaneous polarization switching.

We calculated the Gibbs free energies of hydrogen (H) intermediates ($\Delta G_H$) on various polarized surfaces (Fig. 3i–j). The downward-polarized THZO NWs exhibit a negative Gibbs free energy (−2.74 eV) that favors H atom adsorption but hinders H$_2$ desorption. Conversely, in the upward-polarized state, the positive binding energy (0.21 eV) of H* indicates a weaker interaction with the surface, which enhances H$_2$ desorption.

To further evaluate electron transfer between H and THZO, we conducted charge density difference analyses (Fig. 3k). We observed significant electron density increase around the H atom on the downward-polarized surface, while the H atom exhibited an electron-withdrawing state (blue areas indicate electron consumption, red areas denote charge accumulation). This suggests that electron transfer from THZO to H promotes H* intermediate formation. Additionally, the electron density difference for adjacent O atoms on the downward-polarized surface reinforces how polarization states influence the electron transfer process.

In summary, polarization of the THZO surface modulates the electronic state, promoting covalent bond formation between H and O. After H adsorption, electrons rapidly transfer from the downward-polarized THZO surface to H, resulting in a stable H intermediate. Subsequently, H siphons off another electron through the ferroelectric surface, generating H$_2$. This polarization switch not only initiates free electron flow but also injects electrons into H* during the transition, effectively promoting hydrogen generation. This methodology disrupts the traditional scaling relationship between activation energies for reactant adsorption and product desorption, suggesting potential for efficient piezoelectric water splitting.

We also investigated the decomposition of H$_2$O into ·OH on different polarized surfaces of THZO NWs. As shown in Fig. 3l, m, H$_2$O can stably adsorb on both downward (p↓) and upward (p↑) polarized surfaces, with adsorption energies of −1.23 eV and −0.07 eV, respectively. This indicates strong interaction with both surfaces, though the upward polarized surface is more favorable for H$_2$O adsorption due to its lower energy. The absorbed H$_2$O undergoes cleavage by THZO NWs (H$_2$O* → H* + OH*) to form surface-active groups. The activation energy for OH* on the polarized surface is 0.8 eV, similar to H* adsorption, suggesting that polarization enhances OH* dissociation, allowing more H* to contribute to hydrogen evolution.

Moreover, the planar mean electron density difference around OH* on the downward polarized surface is significantly greater than on the upward surface (Fig. 3n). Thus, the unique polarization characteristics of THZO NWs not only enhance H$_2$O adsorption but also facilitate its activation and dissociation, creating favorable conditions for efficient water decomposition and catalytic performance.

## ROS generation property

The characterization of THZO NWs reveals their capacity for polarization reversal, which enhances piezoelectric catalytic decomposition of water under ultrasonic (US) excitation. To confirm that the subnanometer-scale nanowire structure markedly improves polarization reversal effects compared to THZO NPs, we conducted comparative experiments. As shown in Supplementary Fig. 1, THZO NPs exhibit a uniformly dispersed granular structure, contrasting sharply with the flexible and variable structure of THZO NWs. A comparative analysis of hydrogen (H$_2$) yields from THZO NWs versus THZO NPs in hydrolysis reactions demonstrated that, under US stimulation, THZO NWs significantly reduced the signal intensity of the 2-phenyl-4,4,5,5-tetramethylimidazoline-1-oxyl-3-oxide (PTIO) indicator (Fig. 4a, b and Supplementary Fig. 14). Moreover, THZO NWs exhibits excellent stability in piezoelectric catalytic hydrogen production under multiple ultrasonic repetitions (Supplementary Fig. 15). This reduction suggests

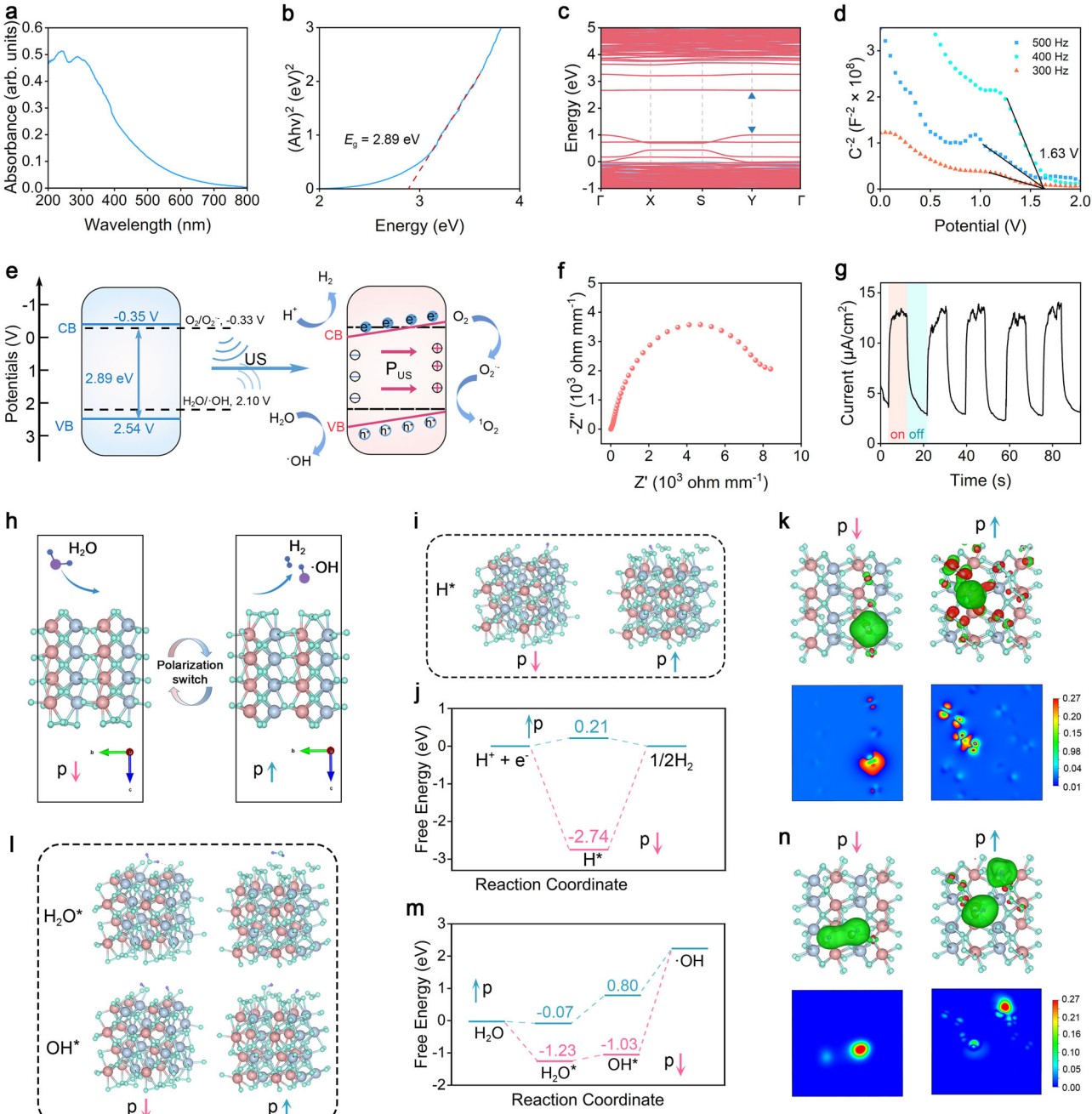

**Fig. 3 | Physical properties of THZO NWs and the H₂O decomposition process of THZO NWs polarization in different directions are proposed by DFT calculation. a** Diffuse absorption spectrum. **b** The bandgap determined using Kubelka−Munk equation. **c** Band structures. **d** The Mott Schottky (M–S) diagrams with different frequency. **d** XPS valence band spectrum. **e** Band structure diagram before and after ultrasonic excitation. **f** Electrochemical impedance spectroscopy (EIS) spectra. **g** Transient piezoelectric current under US irradiation (1 MHz, 0.96 W cm⁻², 40% duty cycle). **h** Schematic diagram of THZO downward (p↓) and upward (p↑) polarization models. **i** H₂ energy maps generated by simulating different polarized surfaces. **j** Simulated surface structure of downward (p↓) and upward (p↑) polarization during catalytic H₂ production. **k** Difference graphs of electron distribution and charge density of H* on different polarized surfaces in THZO NWs. **l** Simulated surface structure of downward (p↓) and upward (p↑) polarization in the process of water decomposition to produce ·OH. **m** Free energy diagram simulating the process of water decomposition to produce ·OH. **n** Difference graphs of electron distribution and charge density of OH* on different polarized surfaces in THZO NWs.

that polarization reversal induced by THZO NWs under US excitation acts as an effective catalyst for accelerating H₂ production.

Monitoring ROS generation using a Rhodamine B (RhB) probe further supported this conclusion (Fig. 4c and Supplementary Figs. 16–17). Notably, THZO NPs have a large band gap of 3.33 eV, which hinders effective electron-hole pair separation and increases resistance to electron transport (Supplementary Fig. 18). Therefore, THZO NWs demonstrate superior water decomposition performance due to

the inversion of iron electrode polarization resulting from their structural deformation. This mechanism challenges the conventional equilibrium between the low activation energy needed for reactant adsorption and the high activation energy required for product desorption on the surface of piezoelectric catalysts, thus paving new avenues for the application of inorganic materials in tumor therapy.

The intense energy of ultrasonic cavitation results in the rapid separation of electron-hole (e⁻–h⁺) pairs. Concurrently, ferroelectric

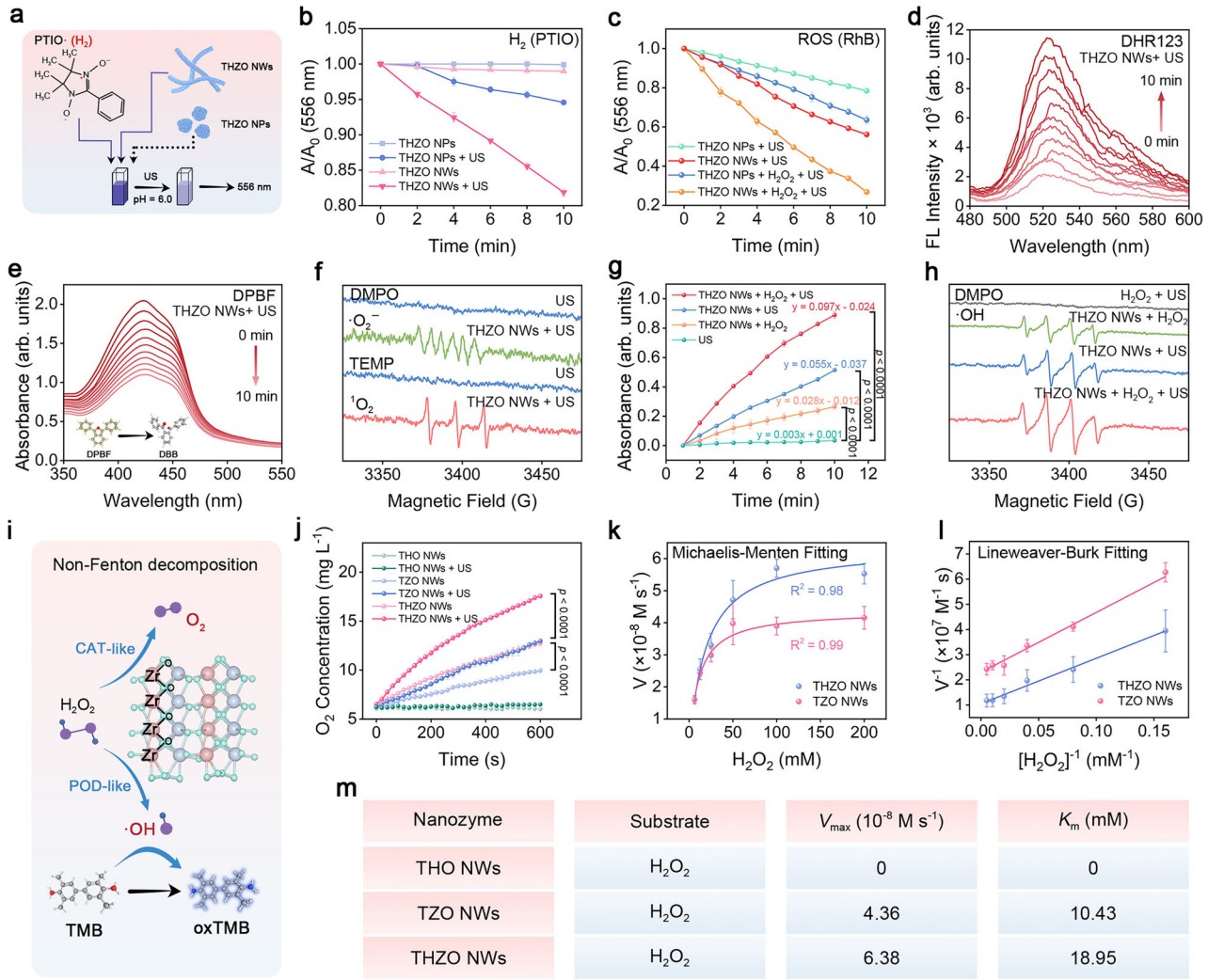

**Fig. 4 | ROS generation property of THZO NWs. a** Schematic diagram of PTIO detecting H$_2$. **b** The relationship between the UV absorption peak intensity of PTIO at 556 nm and the irradiation time under different treatment conditions of THZO NWs and THZO NPs. Data are presented as mean ± S.D. ($n = 3$ independent data). **c** Effect of THZO NWs and THZO NPs on RhB reaction degradation. Data are presented as mean ± S.D. ($n = 3$ independent data). **d** Relationship between DHR123 and irradiation time under THZO NWs + US treatment. **e** Degradation of DPBF as a function of irradiation time under the treatment conditions of THZO NWs + US. **f** ESR spectra of O$_2^-$ trapped by DMPO and $^1$O$_2$ trapped by TEMP of THZO NWs + US.

**g** Absorbance of oxTMB at 652 nm under different conditions over time. Data are presented as mean ± S.D. ($n = 3$ independent data). **h** ESR spectra of ·OH trapped by DMPO in THZO NWs and H$_2$O$_2$ mixed solution. **i** Schematic diagram of enzyme activity reaction of THZO NWs. **j** O$_2$ generation curves of THZO, THO, and TZO NWs under different conditions. Data are presented as mean ± S.D. ($n = 3$ independent data). **k** Michaelis-Menten kinetic analysis and **l** Lineweaver-Burk plot. Data are presented as mean ± S.D. ($n = 3$ independent data). **m** POD-like activity kinetic parameters of THO, TZO, and THZO NWs.

polarization significantly slows the recombination of these pairs, facilitating the reduction of ambient singlet oxygen ($^1$O$_2$) to generate superoxide (O$_2^-$) and the reaction of protons (h$^+$) with water to produce hydroxyl radicals (·OH). The h$^+$ ions subsequently react with O$_2^-$ to form $^1$O$_2$. To confirm that THZO NWs generate O$_2^-$ post-US irradiation, we utilized DHR123 as a fluorescence detection reagent. As shown in Fig. 4d, fluorescence peaks at 520 nm increased with extended US irradiation duration, confirming ROS production under THZO NWs + US conditions. In contrast, no significant change was observed without US application (Supplementary Fig. 19). Additionally, the significant decrease in the peak at 420 nm of 1,3-diphenylisobenzofuran (DPBF) in the presence of p-BQ indicated the production of $^1$O$_2$ (Fig. 4e). Electron spin resonance (ESR) spectroscopy, utilizing DMPO as an O$_2^-$ trapping agent and TEMP as a $^1$O$_2$ trapping agent, verified the ROS species generated (Fig. 4f). We performed sacrificial agent experiments to verify that $^1$O$_2$ was generated by the O$_2$ and ·OH reactions (Supplementary Discussion 2.2 and Supplementary Fig. 20).

We further explored ·OH generation, another hydrolysis product, using TMB as a chromogenic agent. The enhancement of oxTMB absorbance over time suggested a direct correlation between reaction time and ·OH production (Fig. 4g). Notably, TMB showed no changes under US excitation alone (Supplementary Fig. 21a). THZO NWs in the presence of US (Supplementary Fig. 21c) yielded higher ·OH levels than the addition of H$_2$O$_2$ (Supplementary Fig. 21b). The productivity of ·OH under the combined influence of US and H$_2$O$_2$ was determined to be 0.097 min$^{-1}$, while it was 1.028 min$^{-1}$ for pristine THZO NWs + US, ad 0.055 min$^{-1}$ and 0.028 min$^{-1}$ for THZO NWs + H$_2$O$_2$. These represent increases of 1.74 and 3.42 times, respectively (Fig. 4g and Supplementary Fig. 21d).

To further illustrate THZO NWs' capability to generate reactive oxygen species, we utilized 5,5-dimethyl-1-pyrroline N-oxide (DMPO) as a trapping agent for ·OH. The oxidation process of THZO NWs in the catalytic reaction was verified using ESR spectra. As shown in Fig. 4h, the characteristic 1:2:2:1 signal in the ESR spectra of THZO NWs at room

temperature serve as credible evidence for ·OH production. Notably, the peaks in the THZO NWs + H₂O₂ + US group were significantly higher than those in the THZO NWs + US and THZO NWs + H₂O₂ groups, corroborating the results from TMB chromophore detection.

These results demonstrate that THZO NWs possess excellent piezoelectric catalytic properties, with their unique ferroelectric polarization inducing an internal electric field that facilitates not only water molecule splitting but also H₂O₂ decomposition. Importantly, THZO NWs retain a strong ability to decompose H₂O₂ even without US action, prompting further investigation into the underlying principles.

Currently, the traditional understanding of peroxidase (POD)-like and catalase (CAT)-like activities involves electron transfer between substrates and H₂O₂, generating ·OH and O₂. However, THZO NWs (hafnium zirconium oxide nanowires) contain only $Zr^{4+}$ and $Hf^{4+}$ ions, which resist redox reactions yet demonstrate high H₂O₂ decomposition rates. This anomaly arises from a non-Fenton pathway activated on the surface of $ZrO_2$, a transition metal oxide that does not engage in interfacial electron transfer (Fig. 4i). This finding suggests alternative mechanisms for H₂O₂ activation beyond traditional electron transfer processes[39,48]. This reaction mechanism can be divided into three stages:

H₂O₂ undergoes homogeneous cleavage on the surface of $ZrO_2$, generating two ·OH radicals:

$$H_2O_2 \rightarrow 2 \cdot OH \tag{1}$$

These ·OH radicals then interact with H₂O₂, producing $HO_2^{\cdot}$ radicals and water:

$$\cdot OH + H_2O_2 \rightarrow HO_2^{\cdot} + H_2O \tag{2}$$

The accumulated $HO_2^{\cdot}$ radicals in the solution undergo bimolecular decomposition, regenerating H₂O₂ and O₂:

$$2HO_2^{\cdot} \rightarrow H_2O_2 + O_2 \tag{3}$$

Notably, no charged intermediates are involved in the entire reaction process. This non-Fenton pathway for H₂O₂ activation provides a new perspective for understanding the role of transition metal ion oxides in catalytic reactions.

To verify our hypothesis, we synthesized THO NWs and TZO NWs alongside THZO NWs, all featuring similar nanowire architectures (Fig. 4j). Comparative enzymatic activity experiments revealed that TZO NWs and THZO NWs increased oxygen concentration from 6.42 mg L⁻¹ to 9.95 mg L⁻¹ and 12.72 mg L⁻¹, respectively, in the presence of H₂O₂, while THO NWs showed negligible change. Notably, US stimulation further boosted O₂ production to 17.56 mg L⁻¹, highlighting the enhanced CAT-like activity of THZO NWs. Thus, THZO NWs efficiently alleviate TME hypoxia and provide inputs for piezoelectric catalysis. In assays using TMB and H₂O₂, THZO NWs showcased high POD-like activity, with initial reaction rates correlated with H₂O₂ concentration (Supplementary Discussion 2.2, Supplementary Fig. 22 and Fig. 4k–l). The derived $V_{max}$ and $K_m$ values indicated an excellent affinity of THZO NWs for H₂O₂ (Fig. 4m).

In summary, the efficient CAT-like and POD-like enzymatic activities of THZO NWs stem from their effective H₂O₂ decomposition via $ZrO_2$, paving the way for innovative applications in enzymatic therapy and validating their potential as piezoelectric materials for synergistic therapeutic approaches.

## In vitro cellular uptake and cytotoxicity assessment
Encouraged by the impressive multi-enzyme catalytic performance and piezoelectric properties of THZO NWs, we investigated their synergistic anti-tumor effects on the 4T1 cell line in vitro (Fig. 5a). Mitochondria, crucial for aerobic respiration and apoptosis regulation,

are prime drug targets. TPP, a lipophilic cation, effectively delivers bioactive agents to mitochondria. To understand the intracellular behavior of THZO NWs and their relationship with subcellular organelles, we performed fluorescence co-localization experiments using FITC-modified THZO NWs (green) and staining for nuclei and lysosomes with DAPI (blue) and lysosomal tracker (red), respectively (Supplementary Fig. 23). Confocal laser scanning microscopy (CLSM) revealed a significant co-localization of FITC-THZO NWs and lysosomes, with a Pearson's coefficient (PC) of 0.87. Over time, THZO NWs spread throughout the cytoplasm, with the PC decreasing to 0.71 at 2 h and 0.63 at 4 h, indicating efficient lysosomal escape, likely due to the "proton sponge" effect (Fig. 5b).

We then examined the ability of THZO NWs to aggregate in mitochondria post-lysosomal escape (Fig. 5c). Over time, the overlap of green fluorescence from FITC-THZO NWs with red fluorescence (mitochondria) formed yellow/orange fluorescence, with the PC increasing from 0.36 at 1 h to 0.97 at 4 h. Bio-Transmission Electron Microscopy revealed numerous endocytosis vesicles after 1 h of incubation (Fig. 5d–e), indicating strong endocytosis of THZO NWs, likely due to enhanced cellular uptake mechanisms. After 4 h, THZO NWs were observed in mitochondria, which exhibited severe vacuolization, cristae rupture, and cell membrane disruption, typical indicators of apoptosis (Fig. 5f). THZO NWs have low toxicity to normal cells but show effective antitumor activity when combined with US irradiation (Supplementary Discussion 2.3 and Supplementary Fig. 24).

Oxidative stress from increased is critical for tumor cell damage. We assessed intracellular ROS generation using 2,7-dichlorodihydrofluorescein diacetate (DCFH·DA) (Fig. 5h). THZO NWs significantly elevated intracellular ROS levels, with strong green fluorescence in the THZO and THZO + US groups, while control and US-only groups showed weak signals. The fluorescence was notably stronger in the THZO + US group, confirming successful piezoelectric catalysis. Semi-quantitative flow cytometry further supported this, showing the highest ROS levels in the THZO + US group (Supplementary Fig. 25). Dihydroethidium (DHE) analysis also indicated a substantial increase in $O_2^{\cdot-}$ production under piezoelectric catalysis (Supplementary Fig. 26). THZO NWs, especially THZO NWs stimulated by US, destroys mitochondrial function by producing ROS, leading to loss of membrane potential and inducing apoptosis (Supplementary Discussion 2.4 and Supplementary Fig. 27).

To investigate the tumor suppression capability of THZO NWs under US, we employed the cell scratch (Fig. 5i) and colony formation assays (Fig. 5j). The cell scratch assay highlighted reduced migration of 4T1 cells treated with THZO NWs. By 48 h, the control and US groups healed 89.5% and 88.9% of the wound area, respectively, while the THZO group healed only 57.4%. The THZO + US group showed minimal healing, with only a 9.8% increase from 24 h to 48 h, indicating significant inhibition of 4T1 cell migration (Fig. 5k). Colony formation assays supported these findings, showing significant differences in colony formation rates: control (100%), US (98.5%), THZO (47.5%), and THZO + US (12.5%) (Fig. 5l). These results confirm the effective inhibition of migration and proliferation by THZO NWs in combination with US. To visualize treatment effects on cell viability, we performed double staining with Calcein-AM (live cells) and propidium iodide (PI) (dead cells). The THZO + US group exhibited higher red fluorescence intensity, indicating efficient penetration of multicellular tumor spheres and subsequent cell death (Fig. 5m, Supplementary Fig. 28). Flow cytometry revealed an apoptosis rate of 80.3% in the THZO + US group, significantly higher than the control (4.7%) and THZO (53.4%) groups (Fig. 5n–o), corroborating previous MTT assay results.

We explored the specific mechanisms through which THZO NWs induce cell death, hypothesizing that their ability to generate multiple ROS types might trigger pyroptosis, a form of programmed cell death characterized by cell swelling and membrane rupture. Bio-TEM imaging of 4T1 cells post-US irradiation showed numerous vacuoles,

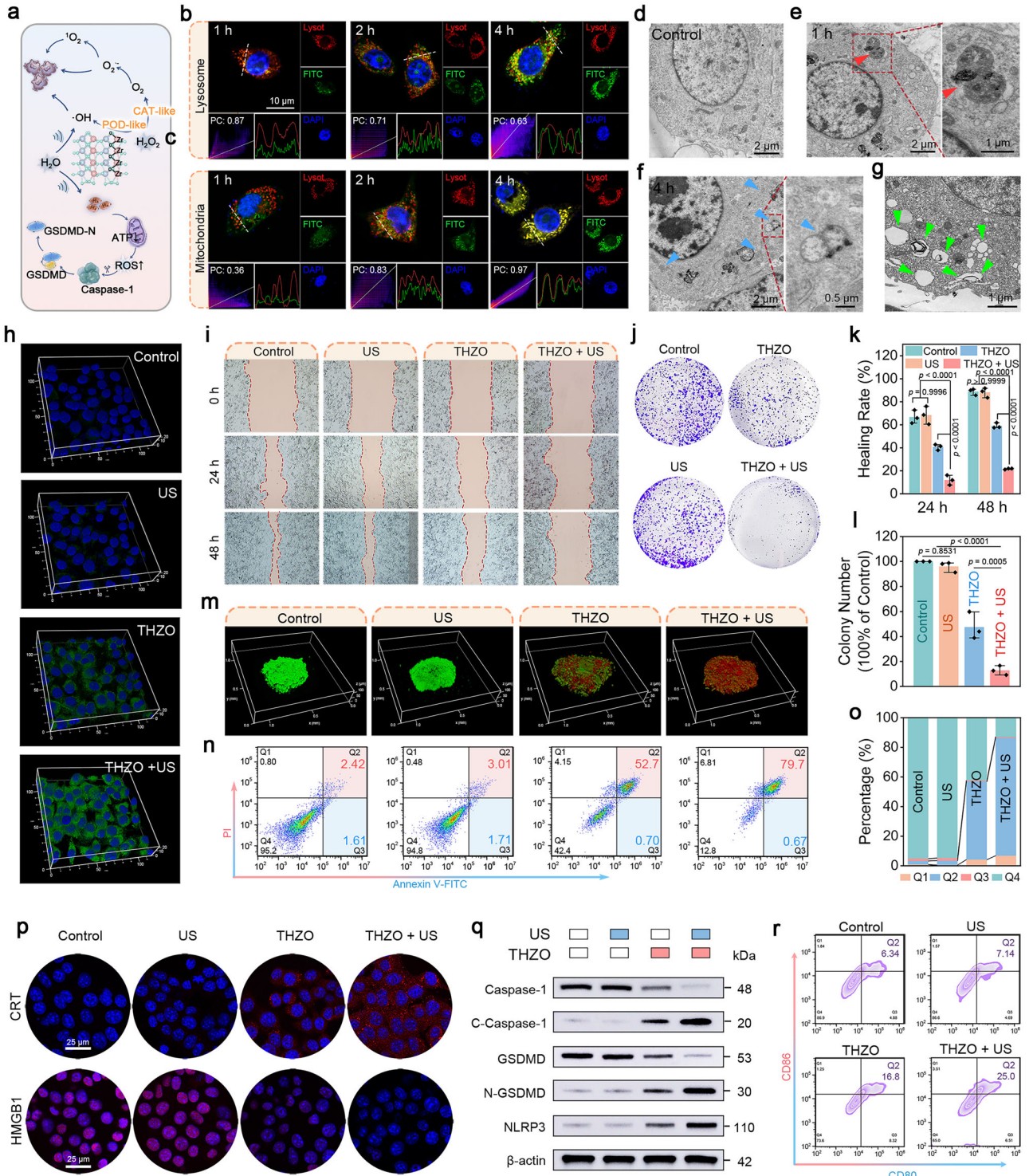

**Fig. 5 | In vitro cellular uptake and pyroptosis. a** Schematic diagram of THZO NWs mediated cooperative piezoelectric catalysis, enzyme catalysis and pyroptosis. CLSM images of FITC-labeled THZO NWs co-located with **b** lysosomes and **c** mitochondria. Bio-TEM images of 4T1 cells incubated with THZO NWs for **d** 0 h, **e** 1 h, and **f** 4 h. **g** Bio-TEM image of 4T1 cells after US treatment. Red and blue arrows locate THZO NWs in lysosomes and mitochondria, respectively. **h** Associated mean fluorescence intensity of intracellular ROS level under various conditions. Blue, DAPI, Green, DCFH-DA. **i** Digital images and **k** corresponding quantitative analysis of 4T1 cell scratching tests. Data are presented as mean ± S.D.

($n$ = 3 biologically independent cell samples). **j** Respective images and **l** corresponding quantification analysis for the colony formation of 4T1 cells upon the indicated treatments. Data are presented as mean ± S.D. ($n$ = 3 biologically independent cell samples). **m** 4T1 multicellular spheroids stained with Calcein-AM/ PI. **n** Flow cytometry analysis of 4T1 cells and **o** corresponding quantitative analysis of cell apoptosis percentages under various conditions. **p** Expression of HMGB1 and CRT in 4T1 cells under different conditions. **q** Western blotting analysis of Caspase-1, C-Caspase-1, GSDMD, N-GSDMD, and NLRP3 expression. **r** Flow cytometry analysis of stimulated maturation state of DCs after different treatments.

indicative of pyroptosis (Fig. 5g). Additionally, we detected $H_2$ levels in cells, with PTIO absorbance decreasing over time, demonstrating THZO NWs' capacity to efficiently produce hydrogen by breaking down water within cells (Supplementary Fig. 29)[49,50]. This hydrogen can diffuse into mitochondria, causing oxidative stress. Although TPP guides nanomaterials near mitochondria, it may inadvertently promote hydrogen and ·OH separation, triggering a unique synergistic mechanism that facilitates tumor cell pyroptosis.

Pyroptosis releases inflammatory factors, activating immune responses characterized by ATP release, calreticulin (CRT) upregulation, and high-mobility group box 1 (HMGB1) secretion[51,52]. ATP levels decreased significantly in the THZO + US group (Supplementary Fig. 30). Immunofluorescence studies showed low CRT signals in control and US groups, contrasted by high fluorescence in the THZO and THZO + US groups. HMGB1 fluorescence intensity decreased dramatically in the THZO + US group (Fig. 5p), indicating facilitated secretion from the nucleus to the cytoplasm.

To investigate the heat death mechanism induced by US in THZO NWs, we conducted Western blotting to analyze key proteins in pyroptosis pathways. In the THZO + US group, cleaved Caspase-1 (C-Caspase-1) levels increased significantly, while Caspase-1 activity decreased (Fig. 5q). Caspase-1 cleaves gasdermin D (GSDMD), releasing its N-terminal domain (N-GSDMD), which forms pores in cell membranes, leading to cell rupture and inflammatory responses marked by NLRP3 inflammasome expression. Subsequently, the results of inflammatory factor detection showed that the expression of inflammatory factors in the THZO NWs + US group was higher, and the expressions of TNF-α and IL-1β were 5.75 and 3.12 times that of the control group, respectively (Supplementary Fig. 31).

Finally, we analyzed the maturation of dendritic cells (DCs) via flow cytometry, observing that THZO NWs significantly increased DC proportions under US irradiation (Fig. 5r). This finding indicates that THZO NWs treated tumor cells effectively activate DCs, promoting anti-tumor immune responses. Thus, THZO NWs combined with US present a promising strategy for enhancing tumor immunity and suppression. High-throughput transcriptome sequencing analysis was performed on 4T1 cells treated in the control group and the THZO NWs + US group. The results showed the activation of pyroptosis caused by $H_2$ and ROS and the core enrichment of most characteristic genes (Supplementary Fig. 32).

## In vivo biodistribution and antitumor effect

Building on the remarkable ferroelectric properties and enhanced dual enzyme activity exhibited by THZONWs in vitro, we further investigated their anti-tumor effects in vivo. To assess CT imaging capabilities, we evaluated the X-ray attenuation coefficients of THZO NWs, as CT is vital for its high resolution and structural detail in medical imaging. Hf, due to its high X-ray absorption coefficient, is a promising contrast agent for in vivo tumor imaging. As shown in Fig. 6a, CT imaging results indicated increased brightness correlating with THZO NW concentration, which significantly enhanced CT signal intensity. A linear relationship between Hounsfield unit (HU) values and THZO NW concentration was established (slope of 42.86 HU $g^{-1}$ $L^{-1}$), providing a basis for quantitative evaluation in CT imaging (Fig. 6b).

To validate THZO NWs' tumor imaging ability in vivo, we intravenously injected THZO NWs into 4T1 tumor-bearing mice and conducted imaging at multiple time points (3, 6, 12, 18, and 24 h). As depicted in Fig. 6c and Supplementary Fig. 33, CT signals at the tumor site significantly increased post-injection, peaking at 12 h, indicating effective tumor contrast enhancement for improved detection accuracy. We also examined the biodistribution of THZO NWs using inductively coupled plasma emission spectroscopy (ICP-OES) after intravenous injection (100 μL, 10 mg $kg^{-1}$). THZO NWs primarily accumulated in the spleen and liver (Fig. 6d), attributed to capture by the reticuloendothelial system. Notably, the elemental Zr content in the

tumor region reached a maximum of 8.75% ID $g^{-1}$ at 12 h, maintaining 4.33% ID $g^{-1}$ at 48 h, highlighting the superior homing ability of THZO NWs to tumors via enhanced permeability and retention effects.

We then explored the anti-tumor effects of THZO NWs in vivo using a subcutaneous 4T1 hormonal Balb/c mouse model. The mice were divided into four groups: control, US only, THZO NWs only, and THZO + US group. A hemolysis test confirmed that THZO NWs ensured safety for intravenous administration, with a hemolysis rate below 1% across all concentrations (Supplementary Fig. 34). During the 14-day treatment, all groups received US irradiation (1.0 MHz, 0.96 W $cm^{-2}$, 3 min, 50% duty cycle) on day 0.5 and day 7.5 (Fig. 6e). The blood half-life of THZO NWs was determined to be biphasic at 0.11 h ($\tau_{1/2}(\alpha)$) and 2.75 h ($\tau_{1/2}(\beta)$) (Fig. 6f), with elimination rate constants of −0.31 and −0.11 μg $mL^{-1}$ (Fig. 6g). These findings suggest favorable circulating and metabolic properties for tumor diagnosis and therapy, effectively reducing toxic side effects while ensuring sufficient drug concentration at the tumor site.

Throughout the 14-day evaluation, body weight changes were monitored, revealing no significant differences among groups, indicating no substantial systemic toxicity from THZO NWs (Fig. 6h). Tumor volume measurements showed significant growth inhibition in THZO and THZO + US groups compared to control and US groups (Fig. 6i, j). The tumor growth inhibition (TGI) rate was highest in the THZO + US group at 96.09%, significantly outperforming the US (2.44%) and THZO (58.44%) groups (Fig. 6i). We corroborated these results by assessing tumor weight and visualizing tumor tissues post-dissection (Fig. 6k, l).

To validate these therapeutic effects histologically, we collected tumor tissues for analysis. TUNEL assays revealed significantly higher fluorescence signals in the THZO + US group, indicating increased cell death (Fig. 6m). H&E staining showed marked cell necrosis and damage in the THZO + US group compared to other groups (Fig. 6m), reinforcing the efficacy of this synergistic treatment strategy. Notably, the THZO + US treatment exhibited superior tumor suppression, highlighting its potential clinical application. Throughout treatment, all mice displayed a slight weight gain, with no apparent side effects, confirming the biosafety of this strategy. Histological analyses of principal organs (heart, liver, spleen, lungs, and kidneys) via H&E staining showed no evident damage, further validating treatment safety (Supplementary Fig. 35). Blood tests conducted on the experimental mouse cohort showed no significant toxicological response in the liver and kidneys following THZO NWs treatment (Supplementary Fig. 36).

Given the effective tumor growth inhibition by THZO NWs in vivo, we investigated the underlying mechanisms, particularly regarding the release of cytoplasmic content that may trigger pyroptosis. Immunofluorescence staining indicated that expression levels of C-Caspase-1, N-GSDMD, and NLRP3 were significantly elevated in tumor tissues treated with THZO NWs, with even greater enhancement in the THZO + US group (Fig. 6m). This confirms the pyroptosis mechanism mediating THZO NWs' anti-tumor immune effects in vivo, consistent with in vitro findings.

To understand the superior performance of THZO NWs in anti-tumor immunostimulation, we quantified dendritic cell (DC) maturation in lymph nodes via flow cytometry. The THZO + US group exhibited significantly higher DC maturation levels than other groups (Fig. 6n, p, Supplementary Fig. 37). Evaluating the spleen for helper T cell ($CD4^+$ T) and cytotoxic T lymphocyte ($CD8^+$ T) activation revealed considerable increases in both populations post THZO + US treatment (29.8% $CD4^+$ and 32.2% $CD8^+$), indicating robust immune response activation (Fig. 6o, q). Furthermore, tumor tissue analysis showed a 5.29-fold increase in $CD8^+$ T cells and a 3.17-fold increase in $CD4^+$ T cells in the THZO + US group compared to controls (Supplementary Fig. 38). Meanwhile, significant tumor suppressive effects can still be observed in immunotherapy combined with PD-1.and the detection

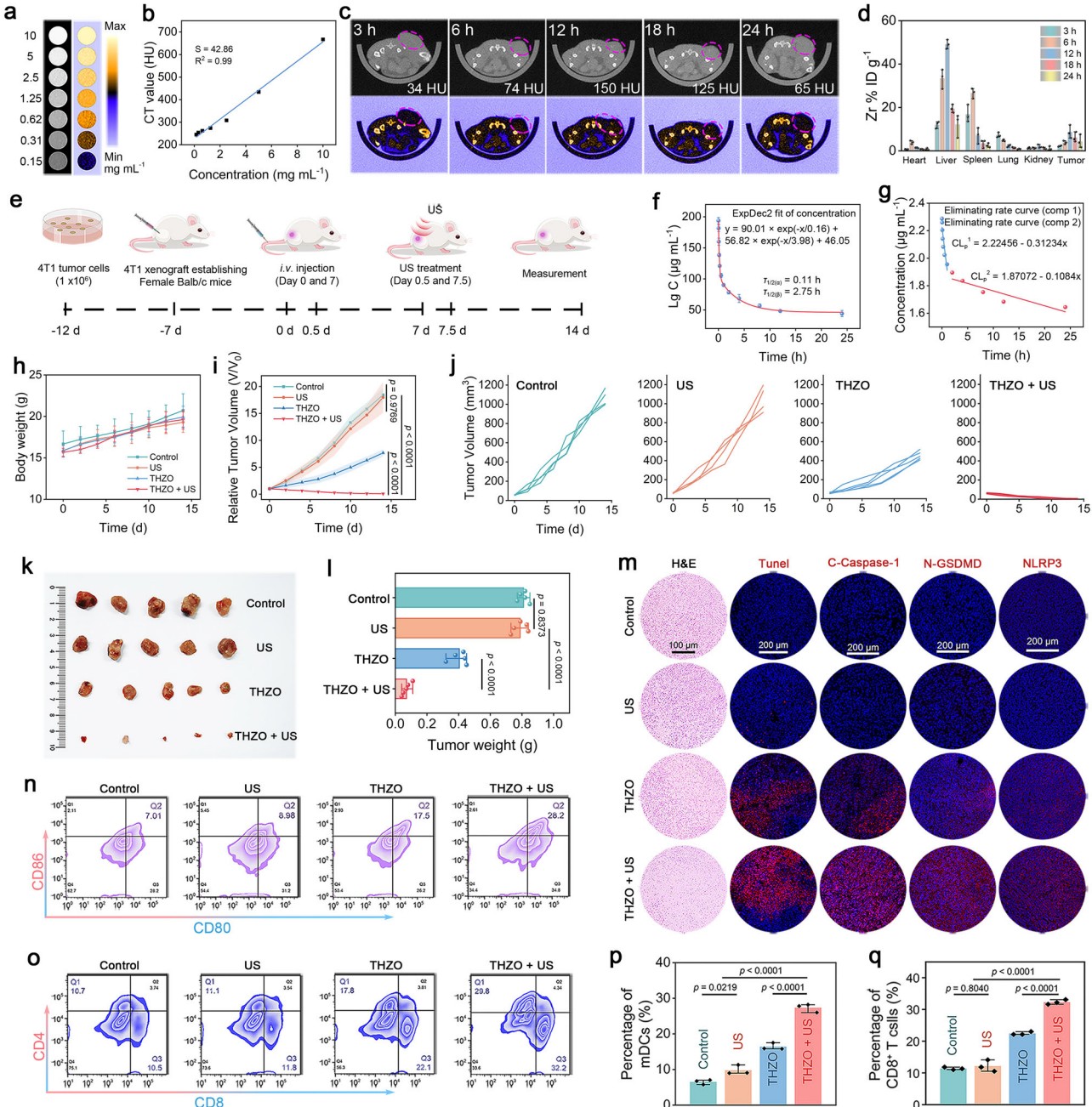

**Fig. 6 | CT imaging and in vivo efficacy of THZO NWs against 4T1 tumor-bearing model. a** In vitro CT pseudocolor images of THZO NWs under different sample concentrations and **b** corresponding CT values versus the concentrations of the samples. **c** In vivo CT images of the tumor site after intravenous administration of THZO NWs at different time intervals. **d** The biodistribution of Zr injected dose (ID) in main tissues and tumors of intravenous administrations of the THZO NWs. Data are presented as mean ± S.D. ($n = 3$ mice). **e** Therapeutic schedule of THZO NWs in 4T1 tumor-bearing mice. **f** The blood circulation curve of intravenously injected THZO NWs. Data are presented as mean ± S.D. ($n = 5$ mice). **g** The eliminating rate curve of intravenously injected THZO NWs from the blood circulation curve. **h** Body weight change and **i** tumor volume change of mice at different days for intravenous injection groups including control, US, THZO, and THZO + US. Data are presented as mean ± S.D. ($n = 5$ mice). **j** Relative tumor growth curves from different groups. **k** Typical tumor photographs excised from the mice after different treatments for 14 days. **l** Average weights of tumors harvested from different groups. Statistical analysis was performed via unpaired two-tailed Student's t-test. ****$p < 0.0001$, n. s., no significance. Data are presented as mean ± S.D. ($n = 5$ one-sided tumor). **m** Histochemical analysis of tumor tissue harvested from mice after various treatments. Representative flow cytometry data of **n** matured DCs and **o** CD8+ T cells in spleens after different treatments. Quantitative analysis of **p** matured DCs and **q** CD8+ T cells in spleens after different treatments. Data are presented as mean ± S.D. ($n = 3$ independent sample).

results of distal tumors showed that THZO + US could significantly inhibit the growth of 4T1 tumors (Supplementary Figs. 39–41).

In conclusion, the significant ROS and $H_2$ production by THZO NWs under ultrasound promotes pyroptosis, resulting in pro-inflammatory cytokine release, DC maturation, and T-cell infiltration, thereby triggering immunostimulation in vivo. This study underscores

the potential of THZO NWs in enhancing tumor immunostimulation through a dual enzyme-ferroelectricity synergistic approach.

## Discussion

We have demonstrated that sub-nanometer THZO NWs exhibit reversible polarization switching under ultrasonic stimulation, leading to a

switchable active surface. Molecular dynamics simulations and DFT calculations substantiate that this phenomenon disrupts the conventional "scaling relationship" and enhances the efficacy of piezoelectric catalysis. We exploited the TME to facilitate the generation of ROS and $H_2$, leveraging the piezoelectric catalysis to achieve effective electron-hole separation. Oxygen gains electrons to form $O_2^{-}$, which then reacts with $h^+$ to produce $^1O_2$. Additionally, THZO NWs exhibit POD-like and CAT-like activities, catalyzing the conversion of $H_2O_2$ to·OH for enzyme therapy and to $O_2$ as a substrate for piezoelectric catalysis. Furthermore, $H_2$ diffusion to mitochondria induces an oxidative stress response, triggering pyroptosis through the canonical ROS/caspase-1/GSDMD pathway, enhancing tumor immunogenicity and activating T cell-mediated adaptive immune responses. These findings offer insights for the design of subnanoscale ferroelectric materials in cancer therapy and explore pyroptosis induced by TME-activated nano-materials in immunostimulation.

## Supplementary methods

### Chemicals and materials
Oleic acid (OA, 90%), oleylamine (OAm, 98%), 1-octadecene (ODE, 95%), Hf(acac)$_4$ (99%), Zr(acac)$_4$ (99%), LiOH (99%), 3,3′,5,5′-tetra-methyl-benzidine (TMB, BR), 1,3-diphenylisobenzofuran (DPBF, 97%), and 5,5-dimethyl-1pyrroline N-oxide (DMPO) were purchased from Sigma-Aldrich. 1,2-Distearoyl-sn-glycero-3-phosphoethanolamine-N-[methoxy(polyethylene glycol)2000] triphenylphosphonium bromide (DSPE-PEG$_{2000}$-TPP, 95%), were obtained from Ruixi Biological Technology Co., Ltd. (Xi'an, China). A variety of biochemical reagents were sourced from Beyotime Institute of Biotechnology, located in Haimen, China. The procured materials included Methyl thiazolyl tetrazolium (MTT, purity exceeding 98%), 2′,7′-Dichlorofluorescein diacetate (DCFH-DA) fluorescein isothiocyanate (FITC, 95% pure), Calcein-AM (with a minimum purity of 97%), propidium iodide (PI, at least 99% pure), along with a JC-1 staining kit and an annexin V-FITC/PI apoptosis detection kit. The PBS and Dulbecco's modified Eagle medium (DMEM) were purchased from Gibco Life Technologies. Beijing Solarbio Science & Technology Co., Ltd. (Beijing, China) provided the Hematoxylin-Eosin (HE) staining reagents employed for the experiments. Anti-CD45-APC (cat.17-0451-82), anti-CD4-FITC (cat.11-0041-82), anti-CD8-PE (cat.12-0081-82), anti-CD11c-APC (cat.17-0114-82), anti-CD80-PE (cat.12-0801-82), and anti-CD86-FITC (cat.11-0862-82) were purchased from Thermo Fisher (USA). Anti-CD3-PerCP/Cya-nine5.5 (cat.100327) was purchased from Biolegend (USA). All chemical reagents in this article were used without further purification.

### Characterization
TEM was performed on an FEI Tecnai T20 instrument to examine morphological features. At the Beamlines MCD-A of NSRL, O k-edge X-ray absorption fine structure (XAFS) spectral data were collected. Atomic-level structural details were investigated with a JEM-ARM200F aberration-corrected electron microscope, which was set at an accelerating voltage of 200 kV. The crystallinity of the synthesized THZO nanowires was characterized using a Rigaku D/MAX-TTR-III X-ray diffractometer with a Cu Kα source, with data collected over a 2θ range from 10° to 80°.Sample absorbance in the ultraviolet-visible region was measured with a UV-1601 spectrophotometer. An ESCALAB 250Xi XPS system was used for XPS measurements. To evaluate surface functional groups on THZO NWs, FTIR spectrometry was carried out on a Perkin-Elmer 580B instrument. ESR spectroscopy using a Bruker EMX1598 spectrometer enabled the identification of reactive oxygen species such as $O_2^{-}$, $^1O_2$, and ·OH. The elemental composition was determined by inductively coupled plasma mass spectrometry (ICP-MS), performed on a Thermo Scientific Icap 6300 instrument. Cellular analyses were performed with a BD Accuri C6 flow cytometer from the United States. Fluorescence imaging was accomplished using a Leica TCS SP8 confocal laser scanning microscope (CLSM).

### PFM
The piezoelectric characteristics of HZO sub-nanowires were evaluated using a standard AFM setup (AIST-NT Smart SPM 1000) under ambient environment conditions, with conductive platinum-coated probes (Mikromasch HQ:NSC35/Pt). The HZO sub-nanowire suspension was treated with multiple washes using heated ethanol before being adjusted to a specific concentration. This suspension was then deposited onto FTO substrates that had been pre-cleaned with ammonia solution (enabling the adhesion of HZO sub-nanowires to the surface via van der Waals interactions or electrostatic forces). Initially, the sample surface was examined in tapping or non-contact mode to locate target particles or nanowires. The system was subsequently switched to Contact Mode, with the setpoint adjusted to apply minimal force. PFM measurements were promptly carried out by applying a low-frequency AC voltage, during which piezoelectric response signals were acquired.

### DFT calculation
The Vienna Ab initio Simulation Package (VASP, Version 6.1.1) was utilized to calculate the Gibbs free energy and DOS for the investigated systems, grounded in density functional theory. To achieve accurate self-consistent determination of the charge density, a plane-wave basis set was applied with a cutoff energy set at 450 eV. Optimization of geometric structures was carried out by enforcing a convergence threshold of 0.001 eV Å$^{-1}$ on the atomic forces. A slab model representing the surface was built with a 30 Å vacuum layer to prevent inter-slab interactions. Sampling of the Brillouin zone during geometry optimizations was conducted using a $2 \times 2 \times 2$ k-point grid (Supplementary Data). Furthermore, weak interactions involving the slabs and reactive species were incorporated via the DFT-D3 dispersion correction scheme.

### Molecular dynamics simulations
The molecular dynamics analyses were performed with the open-source program LAMMPS23. An initial system configuration was generated by positioning a liquid phase inside a bounded simulation cell, facilitated by the PACKMOL package. Interatomic interactions were described through the OPLS-AA force field. All constructed models incorporated periodic boundary conditions. Before the production runs, the starting structures underwent energy minimization via the steepest descent method, and were subsequently equilibrated in a 100 ps molecular dynamics simulation. The simulations adopted the NPT ensemble at 298 K, where the Nose-Hoover method and the Parrinello-Rahman algorithm were utilized to control temperature and pressure, respectively. For visualization and graphical output, the OVITO package was employed.

### Synthesis of Hf$_{0.5}$Zr$_{0.5}$O$_2$ (HZO) NWs
In a typical synthetic procedure, the initial step involved weighing out the reactants: 0.5 mmol of Hf(acac)$_4$, 0.5 mmol of Zr(acac)$_4$, 2 mmol of LiOH, 1 mL of oleic acid (OA), 4.5 mL of oleylamine (OAm), and 10 mL of 1-octadecene (ODE) were placed into a 100 mL of three-necked flask. These components were loaded into a 100 mL three-neck flask. Prior to initiating the reaction, the mixture underwent a degassing treatment at 100 °C under argon protection for 30 min, aiming to eliminate dis-solved oxygen and other potential contaminants. Following this, the reaction system was warmed at a constant rate of 5 °C min$^{-1}$ until the temperature reached 300 °C. After stabilization at the target temperature, the reactants were held under an argon atmosphere for 3 h to ensure the reaction could proceed adequately. Upon completion, the reactants were allowed to cool naturally to room temperature, and the solution containing HZO NWs was collected using a centrifugation technique (11,010 × $g$, 5 min) with ethanol as the solvent. The collected solution was further dispersed in 10 mL of cyclohexane to obtain a stable colloidal solution for subsequent experiments or applications.

## Surface modification of HZO NWs and THZO NWs

For modification of HZO NWs, 1 mL of HZO NWs in cyclohexane with 20 mg of DSPE-PEG-TPP in 10 mL of cyclohexane and stirred continuously for 12 h. Subsequently, in order to remove the cyclohexane solvent, a rotary evaporator is used to evaporate the solution until the solvent is completely removed. Next, the modified THZO NWs were dispersed in deionized water and separated by centrifugation (11,010 × g, 5 min), repeated three times to remove unbound DSPE-PEG-TPP and other possible impurities.

## Synthesis of HZO NPs and THZO NPs

0.5 mmol of Hf(acac)$_4$, 0.5 mmol of Zr(acac)$_4$, 2 mmol LiOH, 1 mL of oleic acid and 4.5 mL of oleylamine were dissolved in 10 mL of diphenyl ether. The reaction mixture was heated to 250 °C with a constant heating rate of 5 °C min$^{-1}$, and then held for 30 min. The resulting solution containing the nanoparticles was then cooled to room temperature. HZO NPs in 1 mL of cyclohexane and DSPE-PEG-TPP in 20 mg cyclohexane were mixed and stirred for 12 h to complete the modification.

## Synthesis of HO NWs and THO NWs

1 mmol of Hf(acac)$_4$, 2 mmol LiOH, 1 mL of oleic acid and 4.5 mL of oleylamine were dissolved in 10 mL of octadecene. The reaction mixture was degassed at 100 °C under argon atmosphere for 30 min. The mixture was heated to 300 °C with a constant heating rate of 5 °C min$^{-1}$, and then held for 3 h under argon atmosphere. The resulting solution containing the NWs was then cooled to room temperature, and 30 mL of ethanol was added to the solution to precipitate the NWs. The NWs were separated by centrifugation (11,010 × g, 5 min) and dispersed in hexane. HO NWs in 1 mL cyclohexane and DSPE-PEG-TPP in 20 mg cyclohexane were mixed and stirred for 12 h to complete the modification.

## Synthesis of ZO NWs and TZO NWs

1 mmol of Zr(acac)$_4$, 2 mmol of LiOH, 1 mL of oleic acid and 4.5 mL of oleylamine were dissolved in 10 mL of octadecene. The reaction mixture was degassed at 100 °C under argon atmosphere for 30 min. The mixture was heated to 300 °C with a constant heating rate of 5 °C min$^{-1}$, and then held for 3 h under argon atmosphere. The obtained mixture of NWs was subsequently brought down to ambient temperature, upon which 30 mL of ethanol was introduced to induce their precipitation. The SNWs were separated by centrifugation (11,010 × g, 5 min) and dispersed in hexane. ZO NWs in 1 mL cyclohexane and DSPE-PEG-TPP in 20 mg cyclohexane were mixed and stirred for 12 h to complete the modification.

## Electrochemistry measurements

The electrochemical characterization of THZO nanowires was carried out with an electrochemical workstation. A 0.5 M Na$_2$SO$_4$ solution served as the electrolyte, while Ag/AgCl and a platinum wire functioned as the reference and counter electrodes, respectively. To prepare the working electrode, a mixture containing 10 mg of the sample dispersed in 0.5 mL of water, 0.5 mL of ethanol, and 32 μL of Nafion (10 mM) was applied and spread onto a nickel foam substrate. The variation of current with time was recorded under the influence of pulsed ultrasound at 40 kHz. Additionally, electrochemical impedance spectroscopy was conducted on the THZO nanowires, and Mott−Schottky curves were acquired at frequencies of 300 Hz and 400 Hz. The flat-band potential relative to the NHE was determined via direct electrochemical measurement and Mott−Schottky analysis, showing that $E_{fb}$ ($vs$ NHE) is approximately equal to $E_{fb}$ ($vs$ Standard Hydrogen Electrode (SHE)). The following section outlines the computational approach used to estimate the energies of the conduction and valence bands. The conversion of the measured flat band potential relative to the Ag/AgCl reference electrode to the value corresponding to the NHE is achieved through the following relationship:

$$E_{fb}(vs\ NHE) = E_{fb}(vs\ Ag/AgCl) + E_{Ag/AgCl} + 0.059 \times pH$$

In this context, $E_{fb}$ ($vs$ NHE) denotes the flat band potential on the NHE scale, which is established under standard conditions of 25 °C using a sodium sulfate electrolyte with a pH of 7. Meanwhile, $E_{fb}$ ($vs$ Ag/AgCl) refers to the experimentally determined flat band potential measured against the Ag/AgCl reference electrode, under identical conditions of 25 °C and a sodium sulfate electrolyte at pH 7.

## Tensile property test

Hydrogels served as the primary matrix, with THZO nanowires acting as performance-enhancing additives. Tensile stress ($\sigma$) was defined as the ratio of the applied load ($F$) to the initial cross-sectional area ($S$) of the sample. The tensile strain ($\varepsilon_l$) was derived from the length ratio $L/L_0$, where $L$ and $L_0$ were the deformed length and original length, respectively. The Young's modulus ($E$) was calculated according to the initial linear slope of the stress−strain curve, while fracture toughness was quantified by computing the total area beneath the stress−strain plot. Additionally, material toughness was evaluated by measuring the area between the loading and unloading curves during cyclic tests.

## ESR measurement

ESR experiments were conducted utilizing the trapping agents DMPO and TEMP. In the case of THZO NWs, a mixture was prepared containing 50 μL of THZO NWs (5 mg mL$^{-1}$), 400 μL of PBS buffer solutions, and 10 μL of either DMPO or TEMP. This mixture was subjected to either US irradiation (1 MHz, 0.96 W cm$^{-2}$, with a 40% duty cycle for 1 min). Subsequently, ·OH radicals, identified by their characteristic 1:2:2:1 signal, and $^1O_2$, identified by their characteristic 1:1:1 signal, were detected using an electron paramagnetic resonance spectrometer.

## Detection of $O_2^{\cdot-}$ and $^1O_2$ in vitro

The concentrations of $O_2^{\cdot-}$ and $^1O_2$ generated by THZO NWs were quantified using DPBF as an indicator. In a standard procedure, a solution containing 50 μg mL$^{-1}$ DPBF, and 100 μg mL$^{-1}$ of THZO NWs was prepared in PBS buffer (pH 6.8). Following a specified reaction period, the absorbance of the solution was measured at 422 nm using a UV-$vis$ spectrophotometer, with the solution being exposed to US irradiation (operating at 1 MHz, 0.96 W cm$^{-2}$, 40% duty cycle for 1 min).

## Detection of ·OH in vitro

The ·OH radicals produced by THZO NWs were quantified using TMB as the substrate. First, a high concentration of TMB dimethyl sulfoxide solution was configured, and subsequent experiments were diluted through PBS buffer. In a standard experiment, a solution was prepared by mixing 400 μg mL$^{-1}$ of TMB and 100 μg mL$^{-1}$ of THZO NWs in PBS buffer (pH 6.8). After allowing the reaction to proceed for a specified duration under US irradiation (operating at 1 MHz, 0.96 W cm$^{-2}$, 40% duty cycle for 1 min), the absorbance of the solution was measured at 652 nm using a UV-$vis$ spectrophotometer.

## Cell culture

The 4T1 murine mammary carcinoma cells, sourced from the Cell Bank, were maintained in RPMI 1640 medium. The culture medium was enriched with 10% fetal bovine serum and 1% penicillin/streptomycin, and the cells were kept at 37 °C in a humidified atmosphere of 5% CO$_2$.

## Cellular uptake experiment

4T1 cells were plated in a culture dish and allowed to proliferate for 24 h until they reached a confluency of ~70−80%. Subsequently, the medium was exchanged with a new one supplemented with THZO nanowires at 100 μg mL$^{-1}$, and the cells were incubated for 1, 2, and 4 h.

At the end of each treatment interval, the cells underwent two washes with PBS before being examined using a CLSM.

## In vitro cytotoxicity assay

4T1 cells were seeded into 96-well plates and incubated for 24 h, and cell density reached ~70–80 %. Following this, the culture medium was replaced with fresh medium containing various concentrations of THZO NWs. For the US-treated groups, the cells were exposed to US irradiation (operating at 1 MHz, 0.96 W cm$^{-2}$, with a 40% duty cycle) for 1 min.

## Detection of intracellular ROS production

4T1 cells were plated in a culture dish and allowed to proliferate for 24 h until they reached a confluency of ~70–80%. Subsequently, the cells were subjected to the following treatments: Control, US (1 MHz, 0.96 W cm$^{-2}$, 40% duty cycle for 1 min), THZO (100 μg mL$^{-1}$), and THZO (100 μg mL$^{-1}$) + US (1 min). After incubation at 37 °C for 4 h, each group was subjected to US treatment. Following this, DCFH-DA was added to the cells at a concentration of 10 μM, and they were incubated in the dark for an additional 40 min. After each treatment, the cells were washed twice with PBS and observed under a CLSM.

## Detection of live/dead cells

4T1 cells were plated in a culture dish and allowed to proliferate for 24 h until they reached a confluency of ~70–80%. Subsequently, they were exposed to the following conditions: Control, US (1 MHz, 0.96 W cm$^{-2}$, 40% duty cycle for 1 min), THZO (100 μg mL$^{-1}$), and THZO (100 μg mL$^{-1}$) + US (1 min). After incubating the cells at 37 °C for 4 h, they were stained with Calcein AM and PI for 30 min. The cells were then washed multiple times with PBS and visualized using CLSM.

## Cell apoptosis analysis

The 4T1 cells were plated in a culture dish and allowed to grow for 24 h, and cell density reached ~70–80 %. Afterward, they were subjected to various treatments: Control, US (1 MHz, 0.96 W cm$^{-2}$, 40% duty cycle for 1 min), THZO (100 μg mL$^{-1}$), and THZO (100 μg mL$^{-1}$) + US (1 min). Following these treatments, the 4T1 cells were stained using the Annexin V-FITC Apoptosis Detection Kit and analyzed by flow cytometry.

## Detection of mitochondrial membrane potential

4T1 cells were seeded in the culture dish and cultured for 24 h, and cell density reached ~70–80 %. Then, 4T1 cells were treated with the following conditions: Control, US (1 MHz, 0.96 W cm$^{-2}$, 40% duty cycle for 1 min), THZO (100 μg mL$^{-1}$), and THZO (100 μg mL$^{-1}$) + US (1 min). Then, 4T1 cells were stained with the mitochondrial membrane potential assay kit with JC-1.

## In vitro tumor spheroid assay

Tumor spheroid models using 4T1 cells were established in 96-well plates pre-coated with a 1.5% agarose layer and allowed to grow until they achieved a diameter of around 600 μm. To evaluate the cytotoxic effects of THZO NWs, spheroids subjected to different treatments were chosen, including: a Control group, an US only group (1 MHz, 0.96 W cm$^{-2}$, 40% duty cycle, 1 min), a THZO-only group (100 μg mL$^{-1}$), and a combined THZO (100 μg mL$^{-1}$) + US (1 min) group. Following treatment, the spheroids were incubated for 1 h with Calcein-AM and PI for live-dead staining. Cellular fluorescence was then visualized by CLSM.

## Western blot assays

4T1 cells were cultured in six-well plates and treated according to the following groups: Control, US (1 MHz, 0.96 W cm$^{-2}$, 40% duty cycle for 1 min), THZO (100 μg mL$^{-1}$), and THZO (100 μg mL$^{-1}$) + US (1 min). After treatment, cells were collected and stained with antibodies NLRP3, GSDMD, N-GSDMD, Caspase-1, and C-Caspase-1.

## CRT and HMGB1 expression analysis

4T1 cells were plated in 6-well plates and, after a 24 h incubation period, were assigned to various treatment groups: Control, US treatment (1 MHz, 0.96 W cm$^{-2}$, 40% duty cycle for 1 min), THZO (100 μg mL$^{-1}$), and THZO (100 μg mL$^{-1}$) + US (1 min). Post-treatment, the 4T1 cells were stained with anti-mouse CRT and HMGB1 antibodies at 37 °C for 1.5 h. They were then labeled with cy5-conjugated secondary antibodies at room temperature for an additional hour. Following a 10 min staining with DAPI, CRT and HMGB1 expression was observed using CLSM.

## Ethical statement

The female BALB/c mice (aged 4 weeks) used in this experiment were procured from Beijing Vital River Laboratory Animal Technology Co., Ltd. (License #1100111084356). The animals were maintained under controlled conditions (23–25 °C, 50 ± 5% humidity, 12 h light/dark cycle) with ad libitum access to food and water. All experimental procedures were approved by the Ethics Committee of the Second Affiliated Hospital of Harbin Medical University and conducted in compliance with the institutional Guidelines for the Care and Use of Laboratory Animals (No. SYDW 2020-051). In accordance with institutional ethics standards, the maximal permitted tumor size (diameter ≤ 17 mm, volume ≤ 1700 mm$^3$) was not exceeded in any of the experiments.

## Animal experiments

In the orthotopic transplantation model of 4T1 breast cancer, $1 \times 10^6$ 4T1 cancer cells suspended in 100 μL were inoculated into the mammary fat pad of female Balb/c mice. Tumor-bearing mice were randomly assigned to four experimental groups ($n = 5$ mice) and received intravenous administrations. The treatment conditions were as follows: a Control group, an Ultrasound (US) treatment group (parameters: 1 MHz, 0.96 W/cm$^2$, 40% duty cycle, 1 min), a THZO group (100 μg mL$^{-1}$), and a combination group receiving both THZO (μg mL$^{-1}$) and US exposure (1 min). Ultrasound was applied on day 0.5 and day 7.5 post-grouping. Body weight and tumor dimensions were measured every 48 h, and tumor volumes were determined using the formula: volume = (length × width$^2$)/2. At the end of the 14-day experimental period, the mice were sacrificed, and tumors were excised. The tumor inhibition rate for each treated group was computed as: rate = $(V_{control} - V_{experiment})/V_{control} \times 100\%$. Furthermore, tumor tissues along with key organs-including the heart, liver, spleen, lungs, and kidneys—were harvested from the various groups, sectioned, and stained with H&E. Additional analysis was performed using TUNEL staining to evaluate apoptotic activity.

The mice bearing tumors were randomly allocated into six distinct groups for the purpose of carrying out the tumor simulation study (Control, US, PD-1, THZO, THZO + US, and THZO + US + PD-1, $n = 5$ mice for each group). On the 0th day, intravenous injections of nanomedicine and PD-1 were administered, and ultrasound therapy was carried out on the 0.5th day. After 2 days of inoculation of 4T1 cells ($1 \times 10^6$ cells) on the right dorsum, the same number of 4T1 cells are inoculated on the left dorsum. Body mass and tumor dimensions were recorded at 48-h intervals. The experiment was terminated on the 15th day through euthanasia of all mice.

## In vivo immunity

Spleen single cell suspension was collected for t cell and DCs activation analysis. The cells were then stained with anti-CD45-APC, anti-CD4-FITC, and anti-CD8-PE, and T cell activation was assessed via flow cytometric analysis. The cells were then stained with anti-CD11c-APC,

anti-CD80-PE, and anti-CD86-FITC antibodies, and the maturation of DCs was analyzed by flow cytometry.

## Statistics and reproducibility

All experimental procedures were conducted independently for a minimum of three replicates. Numerical results are expressed as mean ± standard deviation (mean ± S.D.). For comparisons involving multiple groups, one-way analysis of variance (ANOVA) was applied, whereas differences between two groups were assessed using a two-tailed Student's t-test for unpaired samples. Statistical significance was defined as $p < 0.05$. Both cellular and animal subjects were randomly assigned to their respective experimental groups. Throughout the experimental process and when evaluating results, investigators remained unaware of group assignments.

## Reporting summary

Further information on research design is available in the Nature Portfolio Reporting Summary linked to this article.

## Data availability

All the data generated and analyzed from this study are presented in the article and its Supplementary Information files. The data are also available from the corresponding author upon request. Source data are provided with this paper.

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

## Acknowledgements

Financial support from the National Natural Science Foundation of China (NSFC U22A20347, 52372266, U20A20377, and 51972075), Natural Science Foundation of Shandong Province (ZR2019ZD29), Heilongjiang Provincial Natural Science Foundation of China (ZD2019E004), and the Fundamental Research Funds for the Central Universities are greatly acknowledged. The authors would like to thank Ruifang Shen from Harbin Institute of Technology, Institute of Space Environment and Materials Science (National Major Science Engineering) for CT imaging tests.

## Author contributions

R.Z. conceived the idea and designed the project. Y.Y. performed the experiments and analyzed the results. B.T. assisted with the experiment design and data analyses. R.Z. and L.Y. wrote and revised the original draft of the manuscript. X.Z. performed the DFT calculations. D.Y. reviewed and edited the manuscript. P.Y. supervised the whole project. All authors discussed the results and commented on the manuscript.

## Competing interests

The authors declare no competing interests.
