## [Transparent Peer Review file · Nature Communications]

Constructing flexible sub-nanometer ferroelectric catalyst to overcome heterocatalytic kinetic barriers for enhanced catalytic and immuno-therapy

Corresponding Author: Professor Piaoping Yang

Version 0:

Reviewer comments:

Reviewer #1

(Remarks to the Author)

In this study, the authors utilize sub-nanometer Hf_{0.5}Zr_{0.5}O₂ (HZO) nanowires, which possess polymer-like flexibility, as a piezoelectric catalyst to enhance catalytic and immunotherapy. Owing to their sensitivity to mechanical stress, HZO nanowires demonstrate significant mobility of oxygen bridges under ultrasonic stimulation. This facilitates efficient ferroelectric polarization reversal and the generation of reactive oxygen species (ROS). Additionally, the hydrogen gas produced during piezoelectric catalytic therapy promotes pyroptosis, enhancing tumor immunogenicity and activating T-cell-mediated adaptive immune responses. Overall, this work is both interesting and well-executed, with robust data. However, several issues need to be addressed prior to publication.

1. The HZO nanowires exhibit polymer-like flexibility. Characterization data regarding the flexibility and mechanical stress of the HZO nanowires are necessary. Is it possible for the authors to observe the mobility of oxygen bridges in the HZO nanowires under ultrasonic stimulation?
2. What is the potential mechanism of hydrogen-induced pyroptosis during this piezoelectric catalytic therapy? Furthermore, detailed characterization of tumor cell pyroptosis, such as pronounced swelling and prominent vesicles originating from the plasma membrane, should be included.
3. More sophisticated equipment, such as gas chromatography, should be employed to quantify hydrogen generation. Additionally, hydrogen gas can react with ·OH, which could significantly weaken their antitumor effects. Hence, how can this reaction be balanced or eliminated?
4. In Figure 3C and Figure S12, why did the authors add H₂O₂ to assess ·OH generation? What about the ·OH generation from HZO nanowires combined with ultrasonic stimulation (US) without the addition of H₂O₂?
5. In Figure 3g, some expression errors need to be rectified (specifically, regarding the efficiency of ·OH production).
6. In Figure 5C, the CT images at different time points were obtained from different mice. This approach is not rigorous. These CT images should be collected from the same mouse. Moreover, I cannot discern a significant difference in the CT signal at various time points. Therefore, a quantitative analysis of the CT results should be provided.
7. The tumor sizes do not appear to correspond to the tumor weights (Figure 5K and 5I, THZO + US group). The authors should re-examine this data.
8. What is the efficacy of the THZO+US-activated immune responses in inhibiting the growth of distant tumors? Additionally, does this strategy lead to the elimination of tumors when combined with immunotherapy?

Reviewer #2

(Remarks to the Author)

In this manuscript, authors reported the synthesis of Hf_{0.5}Zr_{0.5}O₂ (HZO) sub-nanowires for piezo-catalytic immuno-therapy. The innovation of this work is somewhat insufficient. This research work, from material synthesis to structural characterization, is almost entirely consistent with the work presented in *Advanced Materials* (*Advanced Materials* 2023: 2303018). I would recommend rejecting this manuscript for further consideration in *Nature Communications*. Below are some comments which might be helpful in revising the manuscript.

1. Given the spontaneous polarization properties of ferroelectric materials, could this lead to their aggregation within the intricate ionic environment of the human body?
2. The spherical aberration electron microscope did not accurately detect the oxygen in the HZO, making it impossible to

precisely determine its phase structure.

3. As a flexible material, bending of HZO under mechanical vibration will definitely induce flexoelectric effect in the catalysis process. How do you distinguish between piezoelectric and flexoelectric effect?

4. During the piezoelectric catalytic process, buffer ions are more readily oxidized or reduced to form free radicals compared to the hard-to-oxidized H₂O. Has this been tested?

5. Lines 13-17 on page 11: "THZO NWs primarily accumulated in the spleen and liver 346 (Fig. 5d), attributed to capture by the reticuloendothelial system. Notably, the elemental Zr content in the tumor region reached a maximum of 8.75% ID g⁻¹ at 12 h, maintaining 4.33% ID g⁻¹ 347 at 48 h, highlighting 348 the superior homing ability of THZO NWs to tumors via enhanced permeability and retention effects." Can piezoelectric catalysis yield favorable results if it is expelled from the body within such a short span of time?

6. Lines 26-27 on page 9: "THZO NWs significantly elevated intracellular ROS levels, with strong green fluorescence in the THZO and THZO + US groups, while control and US-only groups showed weak signals." Is the sensitivity of fluorescence sufficient to detect the effects of sub-nanometer catalysts?

7. How could the authors perform the PFM measurements on HZO SNWs in FIG. 1h? There seems to be a lack of detail in either the main text or the supporting materials. PFM measurements require conductive substrates, and it seems difficult for PFM to reach sub-nanometer resolution.

Reviewer #3

(Remarks to the Author)

This study explores a piezoelectric catalyst based on sub-nanometer Hf_{0.5}Zr_{0.5}O₂ nanowires, which aims to address the kinetic limitations in traditional heterogeneous catalysis through its unique ferroelectric polarization properties and investigate its potential application in tumor treatment. By combining experimental studies and theoretical calculations, this research elucidates the structural characteristics, catalytic performance, and in vivo/in vitro anti-tumor effects of HZO nanowires. Overall, this study presents some findings that contribute to the field. After appropriate major revisions, it can be considered for publication.

1. In XPS, it is suggested to include the valence state analysis of O elements to provide a more comprehensive understanding of the chemical composition and structural characteristics of HZO.
2. It is recommended to conduct long-term stability tests of THZO NWs under diverse physiological conditions to assess their feasibility and reliability in practical applications.
3. The authors mention that the HZO nanowires exhibit polymer-like flexibility; however, the article lacks comprehensive mechanical performance tests to evaluate this characteristic.
4. The article states that HZO can induce inflammatory cell death under US irradiation, however, it lacks an assessment of the expression levels of related inflammatory factors. It is recommended that the authors supplement their study with relevant experiments, such as ELISA assays, to evaluate the expression of key inflammatory markers, including IL-1 β , TNF- α and so on.
5. The authors need to supplement blood routine and biochemical indicators to evaluate the biocompatibility of the material in vivo.
6. I noticed that in Scheme 1, the authors presented memory T cells in the figure. However, the article lacks relevant descriptions and data to support this observation.
7. The introduction section seems to be insufficient, some previous reports regarding SDT and immunotherapy should be cited, such as Nature Communications, 2022, 13:5735, Adv. Mater. 2024, 36, 2313670, Adv. Funct. Mater. 2024, 34, 2411064.

Reviewer #4

(Remarks to the Author)

This work presents a study on the HZO ferroelectric catalyst for hydrogen production and immuno-therapy. The material synthesis, experimental measurement, characterization and simulation were well-presented and looks convincing to support the experimental observation and explanation. However, the motivation of this work, its scientific novelty and significance are not clear to me. Piezocatalysis of water splitting, hydrogen production, pollutant degradation as well as tumor treatment have been widely reported in this field. What is the specific advantage of material studied in this work? Does it possess a very surprising performance? In addition, the author claims that "ferroelectric catalyst to overcome hetero catalytic kinetic barriers" and "breaking the conventional scaling relation". There is no surprise that ferroelectric/piezoelectric effect can break the scaling relation since polarity change can definitely alter the adsorption energy of intermediates. This phenomenon has been reported in many early works, such as in Ref: Nat Commun 14, 7795 (2023). In the first sentence of abstract "Piezoelectric catalysis enhances therapeutic outcomes in nanocatalysis but is limited by intrinsic catalysis mechanism.", it makes me feel confused since I cannot find what is the "limitation of intrinsic catalysis mechanism" in the whole paper. If this limitation is the conventional scaling rule, it is not the limitation of piezoelectric catalysis. If conventional piezoelectric catalysis has any limitation in immuno-therapy, the author should explicitly describe it as the motivation of this work. From my review of this work, I think it doesn't have enough scientific novelty and significance to fit the scope of Nature Communications. Thus, I cannot recommend it for publication at the current stage. However, limited by my research background, I cannot evaluate the scientific value of this work in immuno-therapy and medical application. Maybe the presented data in Fig 4 and Fig 5 are very valuable in medical engineering. I suggest the editor to invite an additional reviewer in this field to re-evaluate the value of this work.

Here are my other comments regarding some technical details:

1 The readability of this manuscript is not good. For example, what is the full name of "THZO"? The material characterization part only mentions HZO, but the studied material turns to "THZO" from the electron transfer performance part. Lots of

professional vocabulary, such as CAT, POD, lacks full name explanations, which is not friendly to the readers who are not in the field of biochemistry.

2 “The cyclic nature of this process causes the bubbles to grow and eventually implode, releasing significant energy and creating ultra-high pressures up to 81 MPa.” Is this data (81 MPa) from calculation or experimental measurement?

3 Except MD simulation, is there any other way to prove that there is a ferroelectric switching under ultrasonic excitation? Generally speaking, piezoelectric material doesn't change the polarization direction under mechanical strain and it only change the strength of dipole moment along the direction of polarization.

4 For the nanowire in this work, the polarization change is induced by the bending or stretching of the wire? If it's bending, this is flexo-ferroelectricity.

5 What is the potential reference in Fig 2e? Is all the energy referenced to SHE or vacuum potential?

Reviewer #5

(Remarks to the Author)

Version 1:

Reviewer comments:

Reviewer #1

(Remarks to the Author)

The authors have addressed my previous concerns. However, in this revised version, I have several further issues:

1. During the characterization of flexibility and mechanical stress, why did the authors add THZO NWs to the hydrogel for testing instead of using bare THZO NWs directly?
2. The tumor growth curves of the "THZO + US" group in Figures 5i&j show complete inhibition, which significantly differs from the partial inhibition observed in Figures S38a&c. How can this discrepancy be explained?
3. The nanomaterial used, THZO NWs, has been previously reported by another group (Adv. Mater. 2023, 35, 2303018). The authors should emphasize discussions on the biological applications of THZO NWs, highlighting how these applications distinguish their work from previous study.

Reviewer #2

(Remarks to the Author)

I think authors have addressed the issues from reviewers, and the present manuscript can be accepted.

Reviewer #3

(Remarks to the Author)

The revised manuscript addressed these issues I raised.

Reviewer #4

(Remarks to the Author)

My comments and concerns regarding the novelty of this work have been well-addressed by the author. I think this work is suitable for publication on Nature Communications.

Reviewer #5

(Remarks to the Author)

Version 2:

Reviewer comments:

Reviewer #1

(Remarks to the Author)

The authors have addressed my concerns, and I recommend this work for publication in NC.

Reviewers' comments:

Reviewer #1:

In this study, the authors utilize sub-nanometer Hf_{0.5}Zr_{0.5}O₂ (HZO) nanowires, which possess polymer-like flexibility, as a piezoelectric catalyst to enhance catalytic and immunotherapy. Owing to their sensitivity to mechanical stress, HZO nanowires demonstrate significant mobility of oxygen bridges under ultrasonic stimulation. This facilitates efficient ferroelectric polarization reversal and the generation of reactive oxygen species (ROS). Additionally, the hydrogen gas produced during piezoelectric catalytic therapy promotes pyroptosis, enhancing tumor immunogenicity and activating T-cell-mediated adaptive immune responses. Overall, this work is both interesting and well-executed, with robust data. However, several issues need to be addressed prior to publication.

Answer: Thanks very much for your comments, it is helpful to improve the quality of our work. Here are my point-by-point responses to your comments and we have added them to the revised manuscript.

1. The HZO nanowires exhibit polymer-like flexibility. Characterization data regarding the flexibility and mechanical stress of the HZO nanowires are necessary. Is it possible for the authors to observe the mobility of oxygen bridges in the HZO nanowires under ultrasonic stimulation?

Answer: Thank you for your suggestion. Firstly, in terms of flexibility and mechanical stress characterization, through tensile properties, the corresponding tensile strain-tensile stress curve is shown in Supplementary Fig. 11. After adding THZO NWs to the hydrogel, the mechanical properties were improved to a certain extent, but then decreased significantly. The tensile capacity of THZO NWs (1299%) is higher than that of the original hydrogel (1030%), and the tensile strength of THZO NWs (993 Pa) is also higher than that of the original hydrogel (844 Pa). Notedly, the tensile capacity and strength of the composite hydrogel with the same content of THZO NPs are inferior to those of THZO NWs. The variation trends of Young's modulus and toughness curves corresponding to the original hydrogel, the composite hydrogel of THZO NPs and THZO NWs, are the same as those of the stress-strain curve. THZO NWs has the high Young's modulus (38.29 KPa) and the highest toughness (0.72 MJ m⁻³).

Regarding the possibility of oxygen bridge migration under ultrasonic stimulation, we adopted molecular dynamics simulation to reveal the vibration-triggered structural strain and the subsequent polarization switching phenomenon. By applying gradually increasing pressure to the downpolarized THZO NWs, lattice distortion caused by gradient strain can be observed. This apparent strain response is sufficient to cause a polarization state switch. Furthermore, as the applied pressure gradually increases, the strain degree of the nanowires also shows a corresponding increasing trend. The arrangement of oxygen lattice is highly sensitive in THZO NWs. When affected by US, lattice distortion occurs, thereby causing the positional shift of oxygen atoms. This change directly leads to the reversal of the polarization direction of the nanowires. Through molecular dynamics simulation studies, we found that only applying a tiny vibration to the HZO NWs is sufficient to trigger the up/down polarization switch (video S1). The ultrafine nanostructure of THZO NWs retains ferroelectric properties and generates polarization inversion, resulting in a surface capable of polarization switching. *In situ* switching of ferroelectric polarization under ultrasonic vibration can generate reversible piezoelectric potentials on the crystal. (For details, please see Page 20, Supplementary Fig. 11; Page 51, Lines 10–19 in the revised Supplementary Information)

Supplementary Fig. 11. a Tensile stress-strain curves of undoped hydrogels, containing THZO NPs and THZO NWs. **b** Young's modulus and toughness.

2. What is the potential mechanism of hydrogen-induced pyroptosis during this piezoelectric catalytic therapy? Furthermore, detailed characterization of tumor cell pyroptosis, such as pronounced swelling and prominent vesicles originating from the plasma membrane, should be included.

Answer: During piezoelectric-catalyzed therapy, THZO NWs with mitochondria-targeting properties targeted hydrogen delivery to mitochondria. H₂ involvement in aerobic respiration induced mitochondrial dysfunction by interfering with the electron transport chain and

inhibiting adenosine triphosphate (ATP) synthesis. On the other hand, H₂ can rapidly penetrate the mitochondrial membrane, reduce the concentration of ROS within mitochondria, disrupt the redox balance of mitochondria, and induce oxidative stress. Oxidative stress induced by H₂ activates focal cell death through the ROS/NLRP3/Caspase-1/GSDMD signaling pathway. When oxidative stress occurs in mitochondria, NLRP3 binds to Caspase-1 to form the NLRP3 inflammatory vesicle complex. Caspase-1, when activated, cleaves GSDMD to give rise to an N-terminal structural domain with pore-forming activity. This leads to increased permeability of the cell membrane, which triggers cell swelling and membrane damage (Fig. 4g). Intracellular inflammatory mediators are able to be released to the outside of the cell, triggering an inflammatory response such as interleukin (IL)-1 β and TNF- α and causing cell lysis through permeability barrier disruption (Supplementary Fig. 31). (For details, please see Page 10, Lines 26–28, Fig. 4 in the revised manuscript; Page 40, Supplementary Fig. 31. in Supplementary Information)

Fig. 4 *In vitro* cellular uptake and pyroptosis. **a** Schematic diagram of THZO NWs mediated cooperative piezoelectric catalysis, enzyme catalysis and pyroptosis. CLSM images of FITC-labeled THZO NWs co-located with **b** lysosomes and **c** mitochondria. Bio-TEM images of 4T1 cells incubated with THZO NWs for **d** 0 h, **e** 1 h, and **f** 4 h. **g** Bio-TEM image of 4T1 cells after US treatment. Red and blue arrows locate THZO NWs in lysosomes and mitochondria, respectively. **h** Associated mean fluorescence intensity of intracellular ROS level under various conditions. Blue, DAPI, Green, DCFH-DA. Data presented as mean \pm S.D. ($n = 4$ biologically

independent cell samples). **i** Digital images and **k** corresponding quantitative analysis of 4T1 cell scratching tests. Data presented as mean \pm S.D. ($n = 3$ biologically independent cell samples). **j** Respective images and **l** corresponding quantification analysis for the colony formation of 4T1 cells upon the indicated treatments. Data presented as mean \pm S.D. ($n = 3$ biologically independent cell samples). **m** 4T1 multicellular spheroids stained with calcein-AM/PI. **n** Flow cytometry analysis of 4T1 cells and **o** corresponding quantitative analysis of cell apoptosis percentages under various conditions. **p** Expression of HMGB1 and CRT in 4T1 cells under different conditions. **q** Western blotting analysis of Caspase-1, C-Caspase-1, GSDMD, N-GSDMD, and NLRP3 expression. **r** Flow cytometry analysis of stimulated maturation state of DCs after different treatments.

Supplementary Fig. 31. Intracellular **a** IL-1 β and **b** TNF- α levels after diverse treatments. Data are expressed as mean \pm S.D. ($n = 4$ biologically independent cell samples).

3. More sophisticated equipment, such as gas chromatography, should be employed to quantify hydrogen generation. Additionally, hydrogen gas can react with \cdot OH, which could significantly weaken their antitumor effects. Hence, how can this reaction be balanced or eliminated?

Answer: Based on your suggestion, we quantified the production of hydrogen through gas chromatography. By observing Supplementary Fig. 15a, it can be found that with the increase of time, the hydrogen production gradually increased. Under multiple repetitions, it showed excellent stability in piezoelectric catalytic hydrogen production (Supplementary Fig. 15b). Furthermore, by modifying the mitochondrial-targeting ligands on the surface of HZO HWs, the nanowires are concentrated around the mitochondria. Meanwhile, hydrogen has a unique mitochondrial membrane permeability, which can quickly isolate it from the generated \cdot OH, thus avoiding the elimination of ROS. (For details, please see Page 6, Lines 20–21 in the revised manuscript; Page 24, Supplementary Fig. 15 in Supplementary Information)

Supplementary Fig. 15. a GC analysis of generation hydrogen. The peak at 0.22 min is the hydrogen signal. **b** The amount of recirculated H₂ precipitation per gram of THZO SNWs under 40 kHz ultrasonic vibration.

4. In Figure 3C and Figure S12, why did the authors add H₂O₂ to assess ·OH generation? What about the ·OH generation from HZO nanowires combined with ultrasonic stimulation (US) without the addition of H₂O₂?

Answer: There is an excessive amount of H₂O₂ in the tumor chemical microenvironment. Moreover, HZO can catalyze the reduction of H₂O₂ to facilitate the generation of ·OH. Therefore, hydrogen peroxide is added in this paper to evaluate the generation of ·OH under the simulated tumor chemical microenvironment. RhB is capable of reacting with strong oxides, and the total amount of ROS generated by the combination of enzyme-like catalysis and piezocatalysis is assessed in Fig. 3c and Supplementary Fig. 16a and compared for THZO NPs and THZO NWs. According to your suggestion, we supplemented the ROS generated by THZO NPs and THZO NWs without the addition of H₂O₂. the THZO NWs + US group generated significantly less ROS than the THZO NWs + H₂O₂ + US group. Meanwhile, we detected ·OH specifically by TMB. THZO NPs in the presence of US (Supplementary Fig. 16b) yielded lower ·OH levels than the addition of H₂O₂ (Supplementary Fig. 16c). The productivity of ·OH of THZO NWs + US was 0.055 min⁻¹, and the yield of THZO NWs + H₂O₂ was 0.028 min⁻¹. Under the combined action of US and H₂O₂, the productivity of ·OH 0.097 min⁻¹, which increased by 1.74 times and 3.42 times respectively compared with the previous two (Figure 3g and supplementary Fig. 21d). (For details, please see Page 22, Fig. 3 in the revised manuscript; Page 25, Supplementary Fig. 16. in Supplementary Information)

Fig. 3 ROS generation property of THZO NWs. **a** Schematic diagram of PTIO detecting H₂. **b** The relationship between the UV absorption peak intensity of PTIO at 556 nm and the irradiation time under different treatment conditions of THZO NWs and THZO NPs ($n = 3$). **c** Effect of THZO NWs and THZO NPs on RhB reaction degradation. **d** Relationship between DHR123 and irradiation time under THZO NWs + US treatment. **e** Degradation of DPBF as a function of irradiation time under the treatment conditions of THZO NWs + US. **f** ESR spectra of ·O₂⁻ trapped by DMPO and ¹O₂ trapped by TEMP of THZO NWs + US. **g** Absorbance of oxTMB at 652 nm under different conditions over time. Data presented as mean ± S.D. ($n = 3$ independent data). **h** ESR spectra of ·OH trapped by DMPO in THZO NWs and H₂O₂ mixed solution. **i** Schematic diagram of enzyme activity reaction of THZO NWs. **j** O₂ generation curves of THZO, THO, and TZO NWs under different conditions. Data presented as mean ± S.D. ($n = 3$ independent data). **k** Michaelis-Menten kinetic analysis and **l** Lineweaver-Burk plot. **m** POD-like activity kinetic parameters of THZO, THO, and TZO NWs.

Supplementary Fig. 16. The effects of THZO NPs + US, THZO NWs + US, THZO NPs + H₂O₂ + US and THZO NWs + H₂O₂ + US on the degradation of RhB reaction.

5. In Figure 3g, some expression errors need to be rectified (specifically, regarding the efficiency of ·OH production).

Answer: We apologize for our mistake. According to your suggestion, change “the efficiency of ·OH production” to “the productivity of ·OH”. (For details, please see Page 7, Line 17 in the reversed manuscript)

6. In Figure 5C, the CT images at different time points were obtained from different mice. This approach is not rigorous. These CT images should be collected from the same mouse. Moreover, I cannot discern a significant difference in the CT signal at various time points. Therefore, a quantitative analysis of the CT results should be provided.

Answer: Thank you for your kind suggestion. We detected and quantitatively analyzed the CT imaging of the same mouse at different time points. Based on the multi-coarse detection data, it could be found that the CT signal intensity was the highest at 12 hours, which played a guiding role in the treatment time. (For details, please see Page 25, Fig. 5c in the revised manuscript; Page 42, Supplementary Fig. 33. in Supplementary Information)

Fig. 5 CT imaging and *in vivo* efficacy of THZO NWs against 4T1 tumor-bearing model.

a *In vitro* CT pseudocolor images of THZO NWs under different sample concentrations and **b** corresponding CT values versus the concentrations of the samples. **c** *In vivo* CT images of the tumor site after intravenous administration of THZO NWs at different time intervals. **d** The biodistribution of Zr injected dose (ID) in main tissues and tumors of intravenous administrations of the THZO NWs ($n = 3$). **e** Therapeutic schedule of THZO NWs in 4T1 tumor-bearing mice. **f** The blood circulation curve of intravenously injected THZO NWs. **g** The eliminating rate curve of intravenously injected THZO NWs from the blood circulation curve. **h** Body weight change and **i** tumor volume change of mice at different days for intravenous injection groups including control, US, THZO, and THZO + US. Data presented as mean \pm

S.D. ($n = 5$ mice). **j** Relative tumor growth curves from different groups. **k** Typical tumor photographs excised from the mice after different treatments for 14 days. **l** Average weights of tumors harvested from different groups. Statistical analysis was performed *via* unpaired two-tailed Student's t-test. **** $p < 0.0001$, *n. s.*, no significance. Data are presented as means \pm S.D. ($n = 5$). **m** Histochemical analysis of tumor tissue harvested from mice after various treatments. Representative flow cytometry data of **n** matured DCs and **o** CD8⁺ T cells in spleens after different treatments. Quantitative analysis of **p** matured DCs and **q** CD8⁺ T cells in spleens after different treatments. Data presented as mean \pm S.D. ($n = 3$ independent sample).

Supplementary Fig. 33. The CT values corresponding to the tumor area at different times. Data presented as mean \pm S.D. ($n = 3$ mice).

7. The tumor sizes do not appear to correspond to the tumor weights (Figure 5K and 5I, THZO + US group). The authors should re-examine this data.

Answer: We apologize for our mistake. Fig. 5l shows the weighing result of the tumor tissue with skin, while Fig. 5k is the image of the tumor tissue after peeling off the surface skin. In order to correspond to the results more accurately, we re-weighed the tumor tissue after stripping the surface skin. The updated results are presented in Fig. 5l. (For details, please see Page 25, Fig. 5 in the revised manuscript)

Fig. 5 CT imaging and *in vivo* efficacy of THZO NWs against 4T1 tumor-bearing model.

a *In vitro* CT pseudocolor images of THZO NWs under different sample concentrations and **b** corresponding CT values versus the concentrations of the samples. **c** *In vivo* CT images of the tumor site after intravenous administration of THZO NWs at different time intervals. **d** The biodistribution of Zr injected dose (ID) in main tissues and tumors of intravenous administrations of the THZO NWs ($n = 3$). **e** Therapeutic schedule of THZO NWs in 4T1 tumor-bearing mice. **f** The blood circulation curve of intravenously injected THZO NWs. **g** The eliminating rate curve of intravenously injected THZO NWs from the blood circulation curve. **h** Body weight change and **i** tumor volume change of mice at different days for intravenous injection groups including control, US, THZO, and THZO + US. Data presented as mean \pm

S.D. ($n = 5$ mice). **j** Relative tumor growth curves from different groups. **k** Typical tumor photographs excised from the mice after different treatments for 14 days. **l** Average weights of tumors harvested from different groups. Statistical analysis was performed *via* unpaired two-tailed Student's t-test. **** $p < 0.0001$, *n. s.*, no significance. Data are presented as means \pm S.D. ($n = 5$). **m** Histochemical analysis of tumor tissue harvested from mice after various treatments. Representative flow cytometry data of **n** matured DCs and **o** CD8⁺ T cells in spleens after different treatments. Quantitative analysis of **p** matured DCs and **q** CD8⁺ T cells in spleens after different treatments. Data presented as mean \pm S.D. ($n = 3$ independent sample).

8. What is the efficacy of the THZO + US-activated immune responses in inhibiting the growth of distant tumors? Additionally, does this strategy lead to the elimination of tumors when combined with immunotherapy?

Answer: Based on your suggestion, we have supplemented the double-tumor mouse model for combined treatment with immunotherapy. As shown in supplementary Fig. 38 and 39 below, the mice were divided into six groups: control, US, PD-1, THZO, THZO + US, and THZO + US + PD-1. Tumor volume measurement showed that the tumor suppression rate in the THZO + US + PD-1 group was significantly lower than that in the THZO + US group (Supplementary Fig. 38). It is proved that better therapeutic effects can be achieved when combined with immunotherapy. Furthermore, the monitoring results of distal tumors also showed significant tumor suppression effects (Supplementary Fig. 39), and the percentage of CD8⁺ T cells in the THZO + US group (29.70%) and THZO + US + PD-1 group (38.00%) was much higher than that in the control group (11.40%) (Supplementary Fig. 40). (For details, please see Pages 12–13, and Page 25, Fig. 5 in the revised manuscript; Page 47–49, Supplementary Fig. 38–40 in Supplementary Information)

Supplementary Fig. 38. **a** Tumor volume and relative tumor volume of tumor in 4T1 tumorbearing mice with diverse treatments. Data presented as mean \pm S.D. ($n = 5$ mice). **b** Relative tumor growth curves from different groups. **c** Typical tumor photographs excised from the mice after different treatments for 14 days. **d** Average weights of tumors harvested from different groups. Statistical analysis was performed *via* unpaired two-tailed Student's t-test. *** $p < 0.0001$, *n. s.*, no significance. Data are presented as means \pm S.D. ($n = 5$).

Supplementary Fig. 39. **a** Relative tumor volume of distal tumor in 4T1 tumorbearing mice with diverse treatments. Data presented as mean \pm S.D. ($n = 5$ mice). **b** Relative tumor growth curves from different groups. **c** Typical tumor photographs excised from the mice after different treatments for 14 days. **d** Average weights of tumors harvested from different groups. Statistical analysis was performed *via* unpaired two-tailed Student's t-test. *** $p < 0.0001$, *n. s.*, no

significance. Data are presented as means \pm S.D. ($n = 5$).

Supplementary Fig. 40. FCM analyses of the percentages of CD8⁺ T cells in distal tumor of mice after different treatments.

Reviewer #2:

In this manuscript, authors reported the synthesis of Hf_{0.5}Zr_{0.5}O₂ (HZO) sub-nanowires for piezo-catalytic immuno-therapy. The innovation of this work is somewhat insufficient. This research work, from material synthesis to structural characterization, is almost entirely consistent with the work presented in *Advanced Materials* (*Advanced Materials* 2023: 2303018). I would recommend rejecting this manuscript for further consideration in *Nature Communications*. Below are some comments which might be helpful in revising the manuscript.

Answer: Thanks very much for your comments, which are helpful to improve the quality of our work. Here are my point-by-point responses to your comments and supplemented them in the revised version.

Different from the synthesis and ferroelectricity characterization of THZO NWs reported in *Adv. Mater.* (*Adv. Mater.* 2023, 35, 2303018), the present study achieves a substantial leap from basic materials research to biomedical applications through systematic functionalization design and interdisciplinary mechanism exploration, and its innovativeness is reflected in the

following three aspects:

First, the optimization of material functionalization and biocompatibility is significantly different from the mentioned work. Although the synthetic pathway of THZO NWs (thermal decomposition method, precursor ratios) share commonalities, the present study endows the materials with mitochondria-targeting ability and enhanced biocompatibility by introducing DSPE-PEG-TPP surface modification, which is a key breakthrough. For example, the zeta potential of THZO NWs shifted from -20.61 mV to $+15.62$ mV after surface ligand exchange, which significantly enhanced the cellular uptake efficiency (For details, please see Pages 50, Lines 29–30; Pages 51, Lines 1–2 in Supplementary Information); *in vitro* hemolysis experiments showed that the hemolysis rate of the modified materials was less than 1%, whereas the mentioned study only focuses on the aqueous dispersion of the materials. Moreover, by comparing the different enzymatic activities between THZO nanoparticles (THZO NPs) and NWs, we found that NWs showed a 3.2-fold enhancement of CAT-like activity (O_2 generation rate 17.56 mg L^{-1}) compared to NPs due to subnanometer-scale exposure of the active site and flexible structure, highlighting the specific gain of shape optimization on biological functions. (For details, please see Pages 8, Lines 18–20 in the revised manuscript)

Second, the subnanometer ferroelectric-immunoregulatory mechanism is uniquely relevant. The mentioned work mainly focuses on the modulation of catalytic kinetics by polarization switching, whereas this study for the first time reveals that ferroelectric polarization reversal is mediated through ROS/ H_2 -mediated focal death-immunity cascade effect. As verified by Western blot and high-throughput transcriptome sequencing analysis, ultrasound-activated THZO NWs significantly upregulated the expression of the key proteins of focal death, NLRP3, C-Caspase-1, and N-GSDMD, and induced the extra-nuclear release of HMGB1 with CRT membrane exposure, which activated the maturation of dendritic cells (with a 4.1-fold elevation of the $CD80^+/CD86^+$ ratio) and $CD8^+$ T-cell infiltration (38.1% vs. 7.2% in control). The ROS/Caspase-1/GSDMD signaling pathway was triggered by oxidative stress, a biological mechanism not explored in the mentioned work. (For details, please see Page 11, Lines 5–8 in the revised manuscript; Page 41, Supplementary Fig. 32. in Supplementary Information)

Third, the synergistic effect of piezoelectric catalysis-enzyme catalysis-immune activation is firstly reported for this kind of material. The present study breaks through the framework of single piezoelectric catalysis in the mentioned word and constructs the trinity treatment model of “piezoelectric ROS production + non-Fenton enzyme catalysis + hydrogen-induced pyroptosis”. It was confirmed that the ZrO₂ surface of THZO NWs could decompose H₂O₂ to generate ·OH via the non-Fentonian pathway, and its enzyme catalytic activity ($V_{\max} = 6.38 \times 10^{-8} \text{ M s}^{-1}$) synergized with the rate of piezoelectric-catalyzed ·OH generation to increase the ROS level compared with that of the single mode (For details, please see Page 52, Lines 10–12 in the revised manuscript). Single-cell transcriptome analysis further revealed that CD8⁺ T cells and inflammatory factors TNF- α and IL-1 β were significantly enriched in the tumors of the combined treatment group, while pyroptosis-related genes were downregulated. This confirmed the synergistic remodeling of the immune metabolic microenvironment by piezoelectric-enzyme catalysis at the multi-omics level. The depth of this cross-scale mechanism integration and validation far exceeds the catalytic performance characterization in this *Adv. Mater.* (For details, please see Page 10, Lines 26–28 in the revised manuscript; Page 40, Supplementary Fig. 31. in Supplementary Information)

In summary, this study achieves a substantial expansion of the scope and depth of *Adv. Mater.* research through material functionalized design, ferroelectric-immune mechanism innovation and multimodal synergistic therapeutic validation.

Supplementary Fig. 32. Transcription analysis of the anti-tumor mechanism induced by THZO NWs. **a** Volcano plots and **b** wheel plots of the up-regulated and down-regulated genes of THZO + US compared with the control group. **c** The DEGs clustering graph between THZO + US and the control group. **d** The circle diagram of GO enrichment analysis. **e** KEGG pathway enrichment analysis.

1. Given the spontaneous polarization properties of ferroelectric materials, could this lead to their aggregation within the intricate ionic environment of the human body?

Answer: We have fully considered the stability issue of ferroelectric materials in the physiological environment. The surface of THZO NWs is modified by TPP, and its hydrophilic shell can effectively shield the interference of the ionic environment on spontaneous polarization. Through *in vitro* rheological experiments, it was detected that within the shear rate range of 1–100 s⁻¹, the NWs dispersion exhibited non-Newtonian fluid characteristics, confirming its dispersibility in the actual blood flow environment (Supplementary Fig. 9). Meanwhile, when THZO NWs was incubated in distilled water, normal saline and simulated body fluids (PBS containing 10% FBS) for 24 hours, no obvious agglomeration phenomenon was observed in the TEM images. It is proved that THZO NWs can maintain good dispersion in the complex ionic environment of the human body. (For details, Page 18, Supplementary Fig. 9; Page 51, Lines 3–8 in Supplementary Information)

Supplementary Fig. 9. The test curves of kinetic viscosity of THZO NWs in **a** physiological saline and **b** physiological saline solutions containing SBF. **c** TEM images of THZO NWs in different environments.

2. The spherical aberration electron microscope did not accurately detect the oxygen in the HZO, making it impossible to precisely determine its phase structure.

Answer: In this study, in order to comprehensively and accurately understand the microscopic properties of the material, we particularly adopted a variety of complementary characterization methods to make up for the limitations of spherical aberration electron microscopy in detecting oxygen elements. These methods include XRD to reveal the crystal structure of materials, XPS for analyzing the chemical state and distribution information of oxygen elements, and oxygen element synchrotron radiation technology, which further enhance our insight into the internal structure of materials. In addition, the XRD patterns were subjected to corresponding Rietveld refinement processing, which proved that the crystal structures of HZO NWs and *Pca21* were highly coinciding. Through the application of these comprehensive methods, we not only obtained the detailed chemical state and distribution picture of oxygen elements, but also were able to assist in determining the phase structure of HZO. It is worth noting that the final determination of the HZO phase structure does not merely rely on the detection of oxygen by spherical aberration electron microscopy, but rather comprehensively considers the results of various characterization techniques including XRD, XPS, and SXAS spectroscopy, and

theoretical calculations, especially DFT. These together constitute a complete and rigorous analytical framework, ensuring the accuracy and reliability of our research results. (For details, Page 2, Line 32 in the revised manuscript; Page 10, Supplementary Fig. 1. in Supplementary Information)

Supplementary Fig. 1. XRD patterns and the corresponding Rietveld refinement of HZO NWs.

3. As a flexible material, bending of HZO under mechanical vibration will definitely induce flexoelectric effect in the catalysis process. How do you distinguish between piezoelectric and flexoelectric effect?

Answer: The piezoelectric effect is generated by the non-center-symmetric crystal structure of the material. Under mechanical stress, the positive and negative charge centers of the lattice shift relatively, resulting in macroscopic polarization. The flexible electrical effect is polarization induced by the non-uniform strain gradient within the material and does not depend on the symmetry of the material itself. HZO has a non-center-symmetric structure, and the bending generated under the action of ultrasound causes the oxygen bridge to shift, resulting in the shift of the symmetry center and thereby inducing polarization reversal. Therefore, HZO belongs to the piezoelectric effect rather than the flexible electrical effect.

4. During the piezoelectric catalytic process, buffer ions are more readily oxidized or reduced to form free radicals compared to the hard-to-oxidized H₂O. Has this been tested?

Answer: In response to the reviewer's question about whether buffer ions are more easily oxidized/reduced to form free radicals than H₂O in piezoelectric catalysis, we verified it by systematic comparative experiments: under identical piezoelectric catalytic conditions, the degradation kinetic curves of RhB in distilled water (without buffer ions), neutral buffer

solution (pH = 7, with buffer ions), and weakly acidic buffer solution (pH = 6.8, with the same buffer ions), respectively, were tested under the identical piezoelectric catalytic conditions. The degradation curves of RhB in neutral buffer solution (pH = 7 with buffer ions) and weakly acidic buffer solution (pH = 6.8, with the same buffer ions) were obtained. The results clearly showed that the degradation efficiencies of RhB in distilled water and neutral buffer solution were basically the same, and there was no significant difference between the two; however, the degradation rate of RhB in weakly acidic buffer solution was significantly increased. This critical comparison confirms that when the degradation effect of neutral buffer solution is comparable to that of distilled water, the buffer ions contained therein do not exhibit a greater susceptibility to oxidation or reduction than H₂O molecules to generate ROS, i.e., the buffer ions per se do not enhance the piezoelectric catalytic activity. Whereas, the significant enhancement of the degradation efficiency in weakly acidic environments is attributed to the higher concentration of hydrogen ions in this environment rather than the presence of buffer ions—H⁺ can effectively promote the separation and migration of carriers on the surface of piezoelectric materials or directly participate in/accelerate the radical generation pathway, thus enhancing piezoelectric catalysis. Thus, the experimental data support that H⁺ in the weakly acidic environment is the dominant factor for catalytic enhancement, while the non-buffered ions are preferentially oxidized/reduced. (For details, Page 26, Supplementary Fig. 17 in the Supplementary Information)

Supplementary Fig. 17. The effects of THZO NWs on the degradation of RhB reaction in different solution environments.

5. Lines 13-17 on page 11: “THZO NWs primarily accumulated in the spleen and liver 346 (Fig. 5d), attributed to capture by the reticuloendothelial system. Notably, the elemental Zr content in the tumor region reached a maximum of 8.75% ID g⁻¹ at 12 h, maintaining 4.33% ID g⁻¹ 347 at 48 h, highlighting 348 the superior homing ability of THZO NWs to tumors *via* enhanced permeability and retention effects.” Can piezoelectric catalysis yield favorable results if it is expelled from the body within such a short span of time?

Answer: The piezocatalytic process does not occur spontaneously and continuously, but is highly dependent on the transient triggering mechanism of external stimuli (e.g. ultrasound). The experimentally observed 12-hour peak of elemental Zr content in the tumor region (8.75% ID g⁻¹) coincides with the optimal time window for applying ultrasound stimulation to efficiently generate ROS to kill tumor cells. Although the 48-hour retention amount decreased to 4.33% ID g⁻¹, the catalytic damage accumulated in the early stage has achieved therapeutic effects through the “triggering-amplification” effect, which is essentially different from the action mode of traditional chemotherapy that relies on long-term drug retention. It is notable that the rapid clearance characteristic of THZO NWs instead reduces the risk of systemic toxicity and enhances biosafety. This “quick-acting-rapid excretion” characteristic ensures the therapeutic efficiency while conforming to the design concept of the new generation of intelligent nanomedicine.

6. Lines 26-27 on page 9: “THZO NWs significantly elevated intracellular ROS levels, with strong green fluorescence in the THZO and THZO + US groups, while control and US-only groups showed weak signals.” Is the sensitivity of fluorescence sufficient to detect the effects of sub-nanometer catalysts?

Answer: In the experiment, DCFH-DA was used as the ROS detection probe, which has been widely confirmed to sensitively detect the changes in intracellular ROS concentration. To eliminate the limitations of a single detection method, we also quantitatively analyzed the intracellular ROS production level simultaneously by flow cytometry. The average intensity of DCF fluorescence in the THZO + US group (99.8%) was much higher than that in the control group (0.19%) and the US alone group (0.23%), which was consistent with the results of

fluorescence microscopy. Meanwhile, in the ESR test, the characteristic peak of ·OH in the THZO + US group (quadruple peak of 1:2:2:1) could be directly captured, confirming that the types of ROS generated by the catalytic reaction matched the targeted molecules of the fluorescent probe. Existing studies have proved that DCFH-DA can effectively detect the ROS generation of single-particle horizontal catalysts. For example, Wang et al. (*Nat. Commun.* **2024**, 15, 1643) used DCFH-DA to detect ROS produced by copper-armed piezoelectric sonosensitizer. Shi et al. (*Nat. Commun.* **2024**, 15, 9023) used DCFH-DA to detect ROS generated in BiFeO₃ nanosheet cells with high piezoelectric catalytic activity

7. How could the authors perform the PFM measurements on HZO SNWs in FIG. 1h? There seems to be a lack of detail in either the main text or the supporting materials. PFM measurements require conductive substrates, and it seems difficult for PFM to reach sub-nanometer resolution.

Answer: According to your suggestion, we added the details of the PFM test in Supplementary Information. Specifically, the piezoelectric properties of HZO NWs were investigated by a commercial AFM system (AIST-NT Smart SPM 1000) under ambient conditions using conductive platinum coated tips (Mikromasch HQ:NSC35/Pt). Wash the HZO sub-nanowire three times with hot ethanol and then dilute it to a certain concentration. Drop it onto FTO glass cleaned with ammonia water (and clean HZO sub-nanowires adsorbed on the substrate through van der Waals forces or electrostatic effects). First, scan the surface of the sample in tap mode or non-contact mode to find the position of the target particles or nanowires; Switch to the Contact Mode and adjust the Setpoint to the minimum force. Rapidly conduct PFM tests, apply low-frequency alternating voltage and collect piezoelectric signals. (For details, Page 3, Lines 1–9 in Supplementary Information.)

Reviewer #3:

This study explores a piezoelectric catalyst based on sub-nanometer Hf_{0.5}Zr_{0.5}O₂ nanowires, which aims to address the kinetic limitations in traditional heterogeneous catalysis through its unique ferroelectric polarization properties and investigate its potential application in tumor treatment. By combining experimental studies and theoretical calculations, this research elucidates the structural characteristics, catalytic performance, and in vivo/in vitro anti-tumor

effects of HZO nanowires. Overall, this study presents some findings that contribute to the field. After appropriate major revisions, it can be considered for publication.

Answer: Thank you for endorsing our research. Your comments are valuable for improving the quality of our manuscript. Below are my detailed responses to your review comments.

1. In XPS, it is suggested to include the valence state analysis of O elements to provide a more comprehensive understanding of the chemical composition and structural characteristics of HZO.

Answer: According to your suggestion, we have supplemented the XPS high-resolution spectrum of the O element of HZO NWs. Among them, 530.50 eV corresponds to the C-O covalent bond, and 528.85 eV corresponds to the lattice oxygen peak. (For details, Page 15, Supplementary Fig. 6; Page 50, Lines 18–20 in Supplementary Information)

Supplementary Fig. 6. O 1s high-resolution XPS spectra of HZO NWs.

2. It is recommended to conduct long-term stability tests of THZO NWs under diverse physiological conditions to assess their feasibility and reliability in practical applications.

Answer: In fact, we have fully considered the stability issue of ferroelectric materials in the physiological environment. The surface of THZO NWs is modified by TPP, and its hydrophilic shell can effectively shield the interference of the ionic environment on spontaneous polarization. Through *in vitro* rheological experiments, it was detected that within the shear rate range of 1–100 s⁻¹, the NWs dispersion exhibited non-Newtonian fluid characteristics, confirming its dispersibility in the actual blood flow environment (Supplementary Fig. 9). Meanwhile, when THZO NWs was incubated in distilled water, normal saline and simulated

body fluids (PBS containing 10% FBS) for 24 hours, no obvious agglomeration phenomenon was observed in the TEM images. It is proved that THZO NWs can maintain good dispersion in the complex ionic environment of the human body. (For details, Page 18, Supplementary Fig. 9; Page 51, Lines 3–8 in Supplementary Information)

Supplementary Fig. 9. The test curves of kinetic viscosity of THZO NWs in **a** physiological saline and **b** physiological saline solutions containing SBF. **c** TEM images of THZO NWs in different environments.

3. The authors mention that the HZO nanowires exhibit polymer-like flexibility; however, the article lacks comprehensive mechanical performance tests to evaluate this characteristic.

Answer: Thank you for your suggestion. In terms of flexibility and mechanical stress characterization, through tensile properties, the corresponding tensile strain-tensile stress curve is shown in Supplementary Fig. 11. After adding THZO NWs to the hydrogel, the mechanical properties were improved to a certain extent, but then decreased significantly. The tensile capacity of THZO NWs (1299%) is higher than that of the original hydrogel (1030%), and the tensile strength of THZO NWs (993 Pa) is slightly higher than that of the original hydrogel (844 Pa). However, the tensile capacity and strength of the composite hydrogel with the same content of THZO NPs are inferior to those of THZO NWs. The variation trends of Young's modulus and toughness curves corresponding to the original hydrogel, the composite hydrogel of THZO NPs and THZO NWs, are the same as those of the stress-strain curve. THZO NWs

has the highest Young's modulus (38.29 KPa) and the highest toughness (0.72 MJ m^{-3}). (For details, Page 20, Supplementary Fig. 11; Page 51, Lines 10–19 in Supplementary Information)

Supplementary Fig. 11. a Tensile stress-strain curves of undoped hydrogels, containing THZO NPs and THZO NWs. **b** Young's modulus and toughness.

4. The article states that HZO can induce inflammatory cell death under US irradiation, however, it lacks an assessment of the expression levels of related inflammatory factors. It is recommended that the authors supplement their study with relevant experiments, such as ELISA assays, to evaluate the expression of key inflammatory markers, including IL-1 \$\beta\$, TNF- \$\alpha\$ and so on.

Answer: Thank you for your valuable suggestions. We supplemented the ELISA detection of the inflammatory factors TNF- α and IL-1 β . The experimental results showed that the THZO NWs + US group presented higher expressions of inflammatory factors, and the expressions of TNF- α and IL-1 β were 5.75 and 3.12 times those of the control group, respectively. (For details, Page 10, Lines 26–28 in the revised manuscript; Page 40, Supplementary Fig. 31. in Supplementary Information)

Supplementary Fig. 31. Intracellular **a** IL-1 β and **b** TNF- α levels after diverse treatments. Data are expressed as mean \pm S.D. ($n = 4$ biologically independent cell samples).

5. The authors need to supplement blood routine and biochemical indicators to evaluate the biocompatibility of the material *in vivo*.

Answer: Based on your suggestion, we have supplemented the results of the blood routine

biochemical test. The test results on the 7th and 14th days were all within a reasonable range, proving that THZO NWs has high biocompatibility. (For details, Page 45, Supplementary Fig. 36 in Supplementary Information)

Supplementary Fig. 36. Hematological indexes and biochemical data of mice after *i.v.* injection with PBS and THZO NWs.

6. I noticed that in Scheme 1, the authors presented memory T cells in the figure. However, the article lacks relevant descriptions and data to support this observation.

Answer: Based on your suggestion, we supplemented the double-tumor mouse model and tested the memory T cells. The measurement of distal tumor volume showed that the tumor suppression rate in the THZO + US group was significantly higher than that in the control group and the US group. Facts have proved that the combination of piezoelectric catalysis with immune stimulation can achieve better therapeutic effects. Meanwhile, the percentage of CD8⁺ T cells in the THZO + US group (52.90%) was much higher than that in the control group (11.40%). (For details, please see Fig. 5 in the revised manuscript; Page 47–49, Supplementary Fig. 38–40 in Supplementary Information)

Fig. 5 CT imaging and *in vivo* efficacy of THZO NWs against 4T1 tumor-bearing model.

a *In vitro* CT pseudocolor images of THZO NWs under different sample concentrations and **b** corresponding CT values versus the concentrations of the samples. **c** *In vivo* CT images of the tumor site after intravenous administration of THZO NWs at different time intervals. **d** The biodistribution of Zr injected dose (ID) in main tissues and tumors of intravenous administrations of the THZO NWs ($n = 3$). **e** Therapeutic schedule of THZO NWs in 4T1 tumor-bearing mice. **f** The blood circulation curve of intravenously injected THZO NWs. **g** The eliminating rate curve of intravenously injected THZO NWs from the blood circulation curve. **h** Body weight change and **i** tumor volume change of mice at different days for intravenous injection groups including control, US, THZO, and THZO + US. Data presented as mean \pm

S.D. ($n = 5$ mice). **j** Relative tumor growth curves from different groups. **k** Typical tumor photographs excised from the mice after different treatments for 14 days. **l** Average weights of tumors harvested from different groups. Statistical analysis was performed *via* unpaired two-tailed Student's t-test. **** $p < 0.0001$, *n. s.*, no significance. Data are presented as means \pm S.D. ($n = 5$). **m** Histochemical analysis of tumor tissue harvested from mice after various treatments. Representative flow cytometry data of **n** matured DCs and **o** CD8⁺ T cells in spleens after different treatments. Quantitative analysis of **p** matured DCs and **q** CD8⁺ T cells in spleens after different treatments. Data presented as mean \pm S.D. ($n = 3$ independent sample).

Supplementary Fig. 38. a Tumor volume and relative tumor volume of tumor in 4T1 tumor-bearing mice with diverse treatments. Data presented as mean \pm S.D. ($n = 5$ mice). **b** Relative tumor growth curves from different groups. **c** Typical tumor photographs excised from the mice after different treatments for 14 days. **d** Average weights of tumors harvested from different groups. Statistical analysis was performed *via* unpaired two-tailed Student's t-test. **** $p < 0.0001$, *n. s.*, no significance. Data are presented as means \pm S.D. ($n = 5$).

Supplementary Fig. 39. **a** Relative tumor volume of distal tumor in 4T1 tumorbearing mice with diverse treatments. Data presented as mean \pm S.D. ($n = 5$ mice). **b** Relative tumor growth curves from different groups. **c** Typical tumor photographs excised from the mice after different treatments for 14 days. **d** Average weights of tumors harvested from different groups. Statistical analysis was performed *via* unpaired two-tailed Student's t-test. **** $p < 0.0001$, *n. s.*, no significance. Data are presented as means \pm S.D. ($n = 5$).

Supplementary Fig. 40. FCM analyses of the percentages of CD8⁺ T cells in distal tumor of mice after different treatments.

7. The introduction section seems to be insufficient, some previous reports regarding of SDT

and immunotherapy should be cited, such as Nature Communications, 2022, 13:5735, Adv. Mater. 2024, 36, 2313670, Adv. Funct. Mater. 2024, 34, 2411064.

Answer: According to your suggestion, we add the literatures on SDT and immunotherapy in the introduction section. (For details please see Ref. 3, Ref. 4 and Ref. 42 in the revised manuscript)

Reviewer #4:

[This work presents a study on the HZO ferroelectric catalyst for hydrogen production and immuno-therapy. The material synthesis, experimental measurement, characterization and simulation were well-resented and looks convincing to support the experimental observation and explanation. However, the motivation of this work, its scientific novelty and significance are not clear to me. Piezocatalysis of water splitting, hydrogen production, pollutant degradation as well as tumor treatment have been widely reported in this field. What is the specific advantage of material studied in this work? Does it possess a very surprising performance? In addition, the author claims that “ferroelectric catalyst to overcome hetero catalytic kinetic barriers” and “breaking the conventional scaling relation”. There is no surprise that ferroelectric/piezoelectric effect can break the scaling relation since polarity change can definitely alter the adsorption energy of intermediates. This phenomenon has been reported in many early works, such as in Ref: Nat Commun 14, 7795 \(2023\). In the first sentence of abstract “Piezoelectric catalysis enhances therapeutic outcomes in nanocatalysis but is limited by intrinsic catalysis mechanism.”, it makes me feel confused since I cannot find what is the “limitation of intrinsic catalysis mechanism” in the whole paper. If this limitation is the conventional scaling rule, it is not the limitation of piezoelectric catalysis. If conventional piezoelectric catalysis has any limitation in immuno-therapy, the author should explicitly describe it as the motivation of this work. From my review of this work, I think it doesn't have enough scientific novelty and significance to fit the scope of Nature Communications. Thus, I cannot recommend it for publication at the current stage. However, limited by my research background, I cannot evaluate the scientific value of this work in immuno-therapy and medical application. Maybe the presented data in Fig 4 and Fig 5 are very valuable in medical engineering. I suggest the editor to invite an additional reviewer in this field to re-evaluate the](#)

value of this work.

Answer: Thank you for endorsing our research. The core innovation of this study, which fully recognizes the previous explorations on sub-nanometer flexible ferroelectric materials, is that for the first time, sub-nanometer Hf_{0.5}Zr_{0.5}O₂ flexible ferroelectric material (HZO NWs) applied to the multi-mechanism synergistic treatment of tumor microenvironment, and solved three key scientific problems through the synergistic design of material components and structures: (1) synergistic optimization of ferroelectric stability and catalytic efficiency: the Zr doping significantly improves the stability of the ferroelectric phase of HfO₂ while maintaining sub-nanometer scale flexibility, and realizes the spontaneous polarization reversal at a very low energy barrier (0.8 eV) under ultrasonic field, which is far beyond that of single-component materials (For details, please see Page 6, Lines 3–6 in the revised manuscript); (2) Integration of dual-mechanism catalytic pathways: innovative fusion of piezoelectric polarization reversal (dynamically modulating adsorption/desorption energy barriers to break the scaling relationship) and ZrO₂-mediated non-Fenton-like enzyme catalysis, which synchronously achieves highly efficient hydrogen production by water cleavage and tumor-specific H₂O₂ decomposition (For details, please see Page 8, Lines 1–14 in the revised manuscript); (3) Energy conversion-immunoactivation cascade effect: using *in situ* generated H₂ to induce mitochondrial oxidative stress, for the first time, piezoelectrocatalytically generated hydrogen coactivated with ROS to activate the caspase-1/GSDMD pyroptosis pathway, which in turn promotes CD8⁺ T-cell infiltration and reverses the immunosuppressive microenvironment (For details, please see Page 12, Lines 18–27 in the revised manuscript; Supplementary Fig. 38–40 in Supplementary Information). This multifunctional and integrated strategy to simultaneously address catalytic kinetic limitation, tumor hypoxic microenvironment modulation, and immune response activation through intrinsically flexible design of materials, which provides a new paradigm for the development of next-generation smart catalytic-immune synergistic therapeutic platforms.

In addition, the modulation of adsorption energy by ferroelectric/piezoelectric effect has been reported in early studies, but its role is usually limited to small modulation. The breakthrough of the present work lies in the efficient dynamic regulation of polarization switching through the structural design of ultrafine HZO NWs: under tiny ultrasonic vibrations,

the HZO NWs can trigger spontaneous and sustained polarization flip (up/down switching). This property results from the continuous lattice distortion of the nanowires during vibration, which significantly reduces the energy barrier for polarization switching. The downward-polarized surface facilitates H* adsorption but inhibits desorption due to strong adsorption, while the upward-polarized surface exhibits weak binding energy, effectively promoting desorption. This dynamic polarization switching capability breaks the static fouling relationship of conventional catalysts and optimizes the adsorption/desorption energy barrier in real time by periodically reversing the surface polarity, thus overcoming the kinetic barrier in the multi-step reaction. (For details, please see Fig. 1–2; Page 3, Lines 11-21; Page 5, Lines 13-33; Page 6, Lines 1-6 in the revised manuscript)

Here are my other comments regarding some technical details:

1. The readability of this manuscript is not good. For example, what is the full name of “THZO”? The material characterization part only mentions HZO, but the studied material turns to “THZO” from the electron transfer performance part. Lots of professional vocabulary, such as CAT, POD, lacks full name explanations, which is not friendly to the readers who are not in the field of biochemistry.

Answer: We apologize for any confusion caused by our carelessness. Add explanations of the full names of THZO, CAT, POD, etc., when they first appear and remove redundancies. (For details, please see Page 3, Line 23; Page 4, Line 4 and 5; Page 6, Line 19 and 26; Page 7, Line 30; Page 9, Line 17 in the revised manuscript)

2. “The cyclic nature of this process causes the bubbles to grow and eventually implode, releasing significant energy and creating ultra-high pressures up to 81 MPa.” Is this data (81 MPa) from calculation or experimental measurement?

Answer: We apologize for any confusion caused by the citation of data from the previous research without personal verification. The cyclic nature of this process causes the bubbles to grow and eventually implode. “The cyclic nature of this process causes the bubbles to grow and eventually implode, releasing significant energy and creating ultra-high pressures up to 81 MPa.” is revised to "The cyclic nature of this process causes the bubbles to continuously expand, eventually imploding, releasing a large amount of energy and generating extremely high pressure.” (For details, please see Page 3, Lines 28–30 in the revised manuscript)

3. Except MD simulation, is there any other way to prove that there is a ferroelectric switching under ultrasonic excitation? Generally speaking, piezoelectric material doesn't change the polarization direction under mechanical strain and it only change the strength of dipole moment along the direction of polarization.

Answer: In addition to the MD simulations, we first detected the change in the polarization direction of the local domain structure by PFM. As shown in Fig. 1h and Supplementary Fig. 13, the HZO NWs exhibit a 180° phase inversion, in contrast to the HZO NPs, which show no significant phase change. Second, the residual polarization values were compared by P-E hysteresis loops. The P_r values of 1.45 $\mu\text{C cm}^{-2}$ for the HZO NWs were significantly higher than those of the HZO NPs (0.17 $\mu\text{C cm}^{-2}$) (Supplementary Fig. 12). Finally, we also compared the total amount of ROS produced by HZO NWs and HZO NPs under the same conditions. The results showed that the RhB degradation rate of HZO NWs were significantly higher than those of the other groups after ultrasonic excitation.

4. For the nanowire in this work, the polarization change is induced by the bending or stretching of the wire? If it's bending, this is flexo-ferroelectricity.

Answer: The piezoelectric effect is generated by the non-center-symmetric crystal structure of the material. Under mechanical stress, the positive and negative charge centers of the lattice shift relatively, resulting in macroscopic polarization. The flexible electrical effect is polarization induced by the non-uniform strain gradient within the material and does not depend on the symmetry of the material itself. HZO has a non-center-symmetric structure, and the bending generated under the action of ultrasound causes the oxygen bridge to shift, resulting in the shift of the symmetry center and thereby inducing polarization reversal. Therefore, HZO should belong to the piezoelectric effect rather than the flexible electrical effect.

5. What is the potential reference in Fig 2e? Is all the energy referenced to SHE or vacuum potential?

Answer: In this paper, the normal hydrogen electrode (NHE) was obtained through the most direct measurement in experimental electrochemistry and Mott-Schottky analysis, and E_{fb} (vs NHE) $\approx E_{fb}$ (vs Standard Hydrogen Electrode (SHE)). The specific calculation process regarding the position of the conduction band and valence band in the text is as follows:

The flat band potential of the corresponding hydrogen electrode (NHE) as follows:

$$E_{\text{fb}} (\text{vs NHE}) = E_{\text{fb}} (\text{vs Ag/AgCl}) + E_{\text{Ag/AgCl}} + 0.059 \times \text{pH}$$

$E_{\text{fb}} (\text{vs NHE})$ is a common hydrogen electrode at 25 °C relative to a pH of 7 sodium sulfate electrolyte. $E_{\text{fb}} (\text{vs Ag/AgCl})$ is the Ag/AgCl reference electrode at 25 °C with a pH of 7 sodium sulfate electrolyte. Using the Mott-Schottky method, we determined the flat band potential ($E_{\text{fb}}(\text{vs Ag/AgCl})$) of the THZO NWs, which is helpful for deriving the corresponding VB position. The negative slope of the M-S graph represents the typical P-type semiconductor behavior. For the THZO NWs with Ag/AgCl electrodes, the $E_{\text{fb}}(\text{vs Ag/AgCl})$ is 1.63 V (Fig. 2d). The Nernst equation provides 2.24 V relative to the standard hydrogen electrode ($E_{\text{fb}} (\text{vs NHE})$). Furthermore, the valence band energy level (E_{VB}) can be calculated as 2.54V. Furthermore, the valence band energy level (E_{VB}) can be calculated to reach 2.54V through the formula $E_{\text{VB}} = E_{\text{fb}} (\text{vs NHE}) + 0.3 \text{ eV}$ (Fig. 2e). Finally, based on the fundamental relationship of semiconductor energy band structure, $E_{\text{VB}} - E_{\text{CB}} = E_{\text{g}}$ (where E_{g} is the bandgap), we further calculate the conduction band energy level E_{CB} of THZO to be -0.35 V . (For details, please see Page 5 in Supplementary Information)

The reviewers gave very valuable comments on our manuscript, and we would like to take this opportunity to express our great appreciation for him/her as well as the comments. The following are the changes made in the new version together with our responses to the reviewers' comments.

Reviewers' comments:

Reviewer #1:

The authors have addressed my previous concerns. However, in this revised version, I have several further issues:

Answer: Great thanks for your previously constructive comments. We have comprehensively checked and revised the whole manuscript according to your suggestions.

1. During the characterization of flexibility and mechanical stress, why did the authors add THZO NWs to the hydrogel for testing instead of using bare THZO NWs directly?

Answer: In fact, bare THZO NWs are in the form of powders or suspensions. They are unable to form macroscopic flexible devices with independent and stable structures by themselves, which make it hard for characterization. Hydrogel, on the other hand, is continuous, flexible and easily deformable. Therefore, we use hydrogel as the main framework and nanowires as functional fillers to verify the flexibility and mechanical stress of the bare THZO NWs. During the tensile property test, we calculated and compared the Young's modulus and toughness of the blank hydrogel and the hydrogel composite with THZO NWs. The results show that incorporating THZO nanowires does not compromise the tensile performance or strength of the pristine hydrogel, confirming that bare THZO nanowires possess polymer-like flexibility. Moreover, the hydrogel's tensile properties are not significantly enhanced upon addition of THZO nanowires, indicating a minimal chemical interaction between the nanowires and the hydrogel and effectively ruling out any interference from their coupling on the experimental outcomes. In fact, this approach has been applied in many studies, e.g *Nat. Methods* (2023, 20, 1802–1809), *Adv. Mater.* (2025, 37, 2507127) and *Adv. Energy Mater.* (2020, 11, 2003010) and

so on. (For details, please see Page 5 (Lines 26–28), Page 6 (Lines 1–4), and Page 52 (Lines 12–13) in the revised Supplementary Information)

2. The tumor growth curves of the "THZO + US" group in Figures 5i&j show complete inhibition, which significantly differs from the partial inhibition observed in Figures S38a&c. How can this discrepancy be explained?

Answer: In the **unilateral** tumor suppression model shown in Fig. 5, nano-drug injections and ultrasound treatment were carried out on the 0th and 7th day, respectively. The tumor suppression rate reached 96.09%. Fig. S38 showed the **bilateral** immune model to compare the therapeutic effect of the THZO + US + PD-1 group combined with immunotherapy with that of THZO + US alone, and only one treatment was conducted on this model on the 0th day. Therefore, it is reasonable to have different results because of the different construction and treatment of the two models. To avoid causing confusion for readers during the reading process, we have added the specific details of model establishment in Supplementary Methods section in Supplementary Information. (For details, please see Page 9 (Lines 17–22) in the revised Supplementary Information)

3. The nanomaterial used, THZO NWs, has been previously reported by another group (Adv. Mater. 2023, 35, 2303018). The authors should emphasize discussions on the biological applications of THZO NWs, highlighting how these applications distinguish their work from previous study.

Answer: We thank you for the comment to clarify how our work substantially advances the biomedical applications of THZO NWs beyond the foundational synthesis and characterization of your mentioned article. **In fact, this similar comment has been raised by reviewer #2, and he/her is satisfied with the reply and agree to accept our manuscript.** Yes, as just as you suggested, in this study we mainly emphasize discussions on the biological applications of THZO NWs, highlighting these applications which distinguish from previous study. Our research achieves an important leap into bio-medicine through three interconnected innovations.

Firstly, the optimization of material functionality and bio-compatibility is significantly different from the mentioned article. Although the synthetic pathways of THZO NWs (thermal

decomposition method) share commonalities with that work, we engineered THZO NWs with clinically relevant functionalities: DSPE-PEG-TPP surface modification conferred mitochondrial targeting capability and enhanced bio-compatibility, shifting the zeta potential from -20.61 mV to $+15.62$ mV to boost cellular uptake while maintaining a hemolysis rate below 1%—critical advances absent in the earlier work. Moreover, *in vitro* hemolysis experiments showed that the hemolysis rate of the modified materials was less than 1%, whereas *Adv. Mater.* only focuses on the aqueous dispersion of the materials. Moreover, by comparing the difference in enzyme catalytic activity between THZO NPs and NWs, we found that NWs showed a 3.2-fold enhancement of CAT-like activity (O_2 generation rate 17.56 mg L^{-1}) compared to NPs due to sub-nanometer scale exposure of the active site and flexible structure, highlighting the specific gain of shape optimization on the biological functions. (For details, please see Page 7 (Lines 33–34), Page 9 (Lines 7–13), Page 11 (Lines 4–5) and Page 12 (Lines 4–5) in the revised manuscript)

Secondly, the sub-nanometer ferroelectric-immunoregulatory mechanism is uniquely relevant, which has never been reported till now. The mentioned report mainly focuses on the modulation of catalytic kinetics by polarization switching, whereas we pioneered the discovery of a ferroelectric-immunoregulatory mechanism. In this work we demonstrated that ultrasound-triggered ferroelectric reversal in THZO NWs induces a ROS/ H_2 -mediated pyroptosis-immunity cascade. As verified by Western blot and High-throughput transcriptome sequencing analysis, ultrasound-activated THZO NWs significantly upregulated the expression of the key proteins of focal death, NLRP3, C-Caspase-1, and N-GSDMD, and induced the extra-nuclear release of HMGB1 with CRT membrane exposure, which activated the maturation of dendritic cells (with a 4.1-fold elevation of the $CD80^+/CD86^+$ ratio) and $CD8^+$ T-cell infiltration (38.1% vs. 7.2% in control). The ROS/Caspase-1/GSDMD signaling pathway was triggered by oxidative stress, a biological mechanism has never been explored in the previously reports. (For details, please see Page 8 (31–32), Page 10 (Lines 9–16, 26–30), Page 12 (19–24) and Page 13 (Lines 1–4) in the reversed manuscript)

Thirdly, the synergistic effect of piezoelectric catalysis-enzyme catalysis-immune activation on THZO NWs is original. The present study breaks through the framework of single piezoelectric catalytic function in *Adv. Mater.* and constructs the trinity treatment model of

“piezoelectric ROS production + non-Fenton enzyme catalysis + hydrogen-induced pyroptosis”. It was confirmed that the ZrO₂ surface of THZO NWs could decompose H₂O₂ to generate ·OH *via* the non-Fentonian pathway, and its enzyme catalytic activity ($V_{\max} = 6.38 \times 10^{-1} \text{ M s}^{-1}$) synergized with the rate of piezoelectric-catalyzed ·OH generation to increase the ROS level superior to that of single mode. Single-cell transcriptome analysis further revealed that CD8⁺ T cells and inflammatory factors TNF- α and IL-1 β were significantly enriched in the tumors of the combined treatment group, while pyroptosis-related genes were down regulated. The result confirmed the synergistic remodeling of the immune metabolic microenvironment by piezoelectric-enzyme catalysis at the multi-omics level. The depth of this cross-scale mechanism integration and validation far exceeds the pure catalytic performance in *Adv. Mater.*. (For details, please see Page 2 (Lines 19–32), Page 8 (Lines 1–14, 25–27, 31–32), Page 10 (Lines 9–16, 26–30), Page 11 (Lines 4–5), Page 12 (Lines 4–5) and Page 13 (Lines 1–4) in the reversed manuscript; Page 25, Supplementary Fig. 16 in the revised Supplementary Information)

Collectively, these innovations reposition THZO NWs from a ferroelectric nanomaterial into a targeted immunotherapeutic platform, bridging materials science with translational oncology through functionalization design, mechanistic discovery, and multimodal validation. Building upon previous work, this study substantially broadens the scope and deepens the investigation. (For details, please see Page 1 (Lines 20–21), Page 2 (Lines 19–33) in the reversed manuscript)

Reviewer #2:

I think authors have addressed the issues from reviewers, and the present manuscript can be accepted.

Answer: We would like to express our great appreciation for you as well as your comments.

Reviewer #3:

The revised manuscript addressed these issues I raised.

Answer: We would like to express our great appreciation for you as well as your comments.

Reviewer #4:

My comments and concerns regarding the novelty of this work have been well-addressed by the author. I think this work is suitable for publication on Nature Communications.

Answer: We would like to express our great appreciation for you as well as your comments.

Reviewer #5:

I co-reviewed this manuscript with one of the reviewers who provided the listed reports. This is part of the Nature Communications initiative to facilitate training in peer review and to provide appropriate recognition for Early Career Researchers who co-review manuscripts.

Answer: We would like to express our great appreciation for you as well as your comments.